# Direct identification of clinically relevant neoepitopes presented on native human melanoma tissue by mass spectrometry

Michal Bassani-Sternberg[1,†,*], Eva Bräunlein[2,*], Richard Klar[2], Thomas Engleitner[3,4], Pavel Sinitcyn[1], Stefan Audehm[2], Melanie Straub[5], Julia Weber[3,4], Julia Slotta-Huspenina[5,6], Katja Specht[5], Marc E. Martignoni[7], Angelika Werner[7], Rüdiger Hein[8], Dirk H. Busch[9], Christian Peschel[2,4], Roland Rad[3,4], Jürgen Cox[1], Matthias Mann[1,**] & Angela M. Krackhardt[2,4,**]

Although mutations may represent attractive targets for immunotherapy, direct identification of mutated peptide ligands isolated from human leucocyte antigens (HLA) on the surface of native tumour tissue has so far not been successful. Using advanced mass spectrometry (MS) analysis, we survey the melanoma-associated immunopeptidome to a depth of 95,500 patient-presented peptides. We thereby discover a large spectrum of attractive target antigen candidates including cancer testis antigens and phosphopeptides. Most importantly, we identify peptide ligands presented on native tumour tissue samples harbouring somatic mutations. Four of eleven mutated ligands prove to be immunogenic by neoantigen-specific T-cell responses. Moreover, tumour-reactive T cells with specificity for selected neoantigens identified by MS are detected in the patient's tumour and peripheral blood. We conclude that direct identification of mutated peptide ligands from primary tumour material by MS is possible and yields true neoepitopes with high relevance for immunotherapeutic strategies in cancer.

[1] Department of Proteomics and Signal Transduction, Max Planck Institute of Biochemistry, Am Klopferspitz 18, Martinsried 82152, Germany. [2] IIIrd Medical Department, Klinikum rechts der Isar, Technische Universität München, Ismaningerstr. 22, Munich 81675, Germany. [3] IInd Medical Department, Klinikum rechts der Isar, Technische Universität München, Ismaningerstr. 22, Munich 81675, Germany. [4] German Cancer Consortium of Translational Cancer Research (DKTK) and German Cancer Research Center (DKFZ), Heidelberg 69120, Germany. [5] Institute of Pathology, Technische Universität München, Ismaningerstr. 22, Munich 81675, Germany. [6] MRI-TUM-Biobank at the Institute of Pathology, Technische Universität München, Ismaningerstr. 22, Munich 81675, Germany. [7] Surgery Department, Klinikum rechts der Isar, Technische Universität München, Ismaningerstr. 22, Munich, 81675, Germany. [8] Dermatology Department, Klinikum rechts der Isar, Technische Universität München, Biedersteiner Str 29, Munich 80802, Germany. [9] Institute for Medical Microbiology, Immunology and Hygiene, Technische Universität München, Trogerstr. 30, Munich 81675, Germany. † Present address: Department of Oncology, UNIL/CHUV, Ludwig Cancer Research Center, Epalinges 1066, Switzerland. * These authors contributed equally to this work. ** These authors jointly supervised this work. Correspondence and requests for materials should be addressed to M.M. (email: mmann@biochem.mpg.de) or to A.M.K. (email: angela.krackhardt@tum.de).

Cancer immunotherapy has demonstrated remarkable efficacy in a large variety of neoplasms and is currently revolutionizing the treatment of malignant diseases. Immune checkpoint modulation, in particular, is emerging as a highly effective therapeutic strategy in an increasing number of cancer entities[1,2]. To further improve current immunotherapeutic approaches, understanding the nature of immunological tumour recognition is of utmost importance. This may be important also for the identification of suitable biomarkers influencing decisions regarding therapeutic sequences and combinations. A number of tumour-associated antigens (TAA) have been evaluated as target antigens in clinical investigations especially in patients with melanoma. These include antigens derived from differentiation antigens and cancer testis antigens[3,4]. However, so far these approaches showed only limited efficacy. Adoptive transfer of T cells transgenic for T-cell receptors (TCR) specific for selected TAA seem to be a reasonable and effective therapeutic development especially using affinity maturated TCR or ones selected from a non-self environment[5,6]. However, severe or even fatal side effects have been observed[5–8]. Response rates observed following treatment with immune checkpoint inhibitors have demonstrated that effective immune responses can be induced in an autologous environment in a significant proportion of melanoma patients[9–11]. Response rates correlate to the mutational load of patients' tumours as shown for melanoma and lung cancer, demonstrating that neoantigens comprising such mutations play a crucial role in anti-cancer immunoreactivity[12–14]. Cancer genomics allows us to precisely determine the landscape of tumour-specific mutations from which such neoantigens may derive[15]. However, our knowledge about defined and clinically relevant tumour-specific antigens (TSA) presented by human leucocyte antigens (HLA) and recognizable by T cells is still very limited. Most efforts to define such antigens in humans and mice currently employ exome and transcriptome analyses followed by in silico epitope prediction and large-scale immunogenicity assays[16–19]. This approach results in many predicted peptide ligands, only few of which have proven to be immunogenic. Peptide ligands selected for therapeutic targeting by prediction may therefore not be clinically effective. Direct identification of neoantigens by tumour-infiltrating T cells is highly laborious and time-consuming[20], and therefore less suitable for clinical translation. There are few reports about the direct identification of neoantigens by the analysis of the tumour ligandome using mass spectrometry (MS) and subsequent matching with exome and transcriptome data[21–23]. Importantly, this approach resulted in the direct identification of therapeutically relevant TSA in two murine models[21,22]. However, so far mutated peptide ligands identified by MS were derived from analyses of monoclonal cell lines only[21–23], not representing the complex heterogeneity of native tumours. Thereby, especially those clonal mutations representing particular promising target antigens for prolonged tumour rejection[24] may be missed. Direct identification of neoantigens from native tumour tissue samples was so far impeded by limitations in sensitivity and bioinformatics. However, translated to human patients, this would represent a major advance for clinical translation of neoantigen-directed immunotherapies.

We hereby report on the application of our recently developed high sensitivity mass spectrometry workflow[25] to the analysis of 25 human native tumour specimens. We provide an unprecedented depth of the tumour-derived ligandome harbouring a broad spectrum of attractive tumour-associated antigens. Most importantly, we discover tumour-specific neoantigens in selected patients validated by the proof of potent patients' derived neoantigen-specific anti-tumour immune responses. Thus, these data demonstrate that high sensitivity MS is a powerful tool to identify neoantigens highly relevant for the development and optimization of personalized immunotherapies in patients with cancer.

## Results

**In-depth immunopeptidomics on native melanoma tissue samples.** Tumour tissue samples from 25 melanoma patients (Supplementary Tables 1 and 2) were used for analysis of biochemically purified HLA class I and II binding peptides. In total, we performed 140 MS measurements of purified peptides by LC–MS/MS analysis (Supplementary Data 1) using a state-of-the-art mass spectrometer, followed by stringent bio(informatics) analyses in the MaxQuant environment[26]. We identify 95,662 unique peptide sequences with a false discovery rate (FDR) of 1% (Fig. 1a, Supplementary Data 2) and report in total 99,355 peptide forms. We discover 78,605 peptides in the HLA class I peptidome from 12,663 proteins and 15,009 in the HLA class II peptidome from 2,832 proteins. In addition, 2,048 peptides from 746 proteins are detected in both classes I and II peptidome samples. The large variability in the number of eluted peptides per patient is in agreement with the amount of eluted HLA complexes. We demonstrate this by showing significant positive correlation between the number of identified peptides in HLA class I peptidome and the amount of recovered beta-2 microglobulin (B2M) in each tissue (Supplementary Fig. 1a and b). Eluted peptides show the characteristic length distribution and the MS-data itself assigns to proper anchor residues of defined HLA allotypes as exemplarily shown for two patients in Fig. 1b,c using the Gibbs clustering approach[27]. Many of the longer peptides (up to 15 amino acids) identified among the eluted HLA class I peptides still show the typical anchor motifs and are therefore likely binders and not contaminants (Fig. 1b,c). Another common approach used to assess purity and overall performance of elution of HLA peptides is the estimation of the affinity of the eluted peptides to the respective HLA molecules by predicting binding affinities[25,28]. This analysis, however, depends very much on the performance of the prediction programs. We predicted the binding affinity of eluted HLA-I peptides from patients Mel15 and Mel16. Due to the difficulty in assignment of peptides with multiple potential restrictions, we filtered the list of peptides to include only 9-mer peptides that bind to only one defined HLA allotype according to the minimum predicted affinity. Instead of using the 500 nM threshold commonly used for peptide binding prediction, we set the threshold for a binding as rank < 2% (standard settings in NetMHC4.0). Using our dataset, we observed that a considerable amount of peptides that was assigned as HLA-B3503 binders fit the binding motif (Pro and Ala in the second position and Leu in the last position). In contrast, the predicted affinities of these binders (rank < 2%) were extremely low, with median predicted affinity of 2,806 nM ($n = 581$) (Fig. 1d,e). These results differ substantially from analysis of HLA-I peptides assigned to HLA-B0702, an allotype with a rather similar motif, for which we observed a median predicted affinity of 17.7 nM of associated peptides ($n = 1,191$) eluted from the tumour of patient Mel16.

**Peptide ligands derived from tumour-associated antigens.** The depth of the ligandome analysis is demonstrated by the identification of a large number of both known and novel peptide ligands derived from described melanocyte-associated differentiation and cancer testis antigens (Fig. 2; Supplementary Data 3). We detected the highest number of TAA-derived peptide ligands for the well-known melanoma-associated antigen PMEL (gp100), (64 HLA I and 46 HLA II ligands) a few of which were

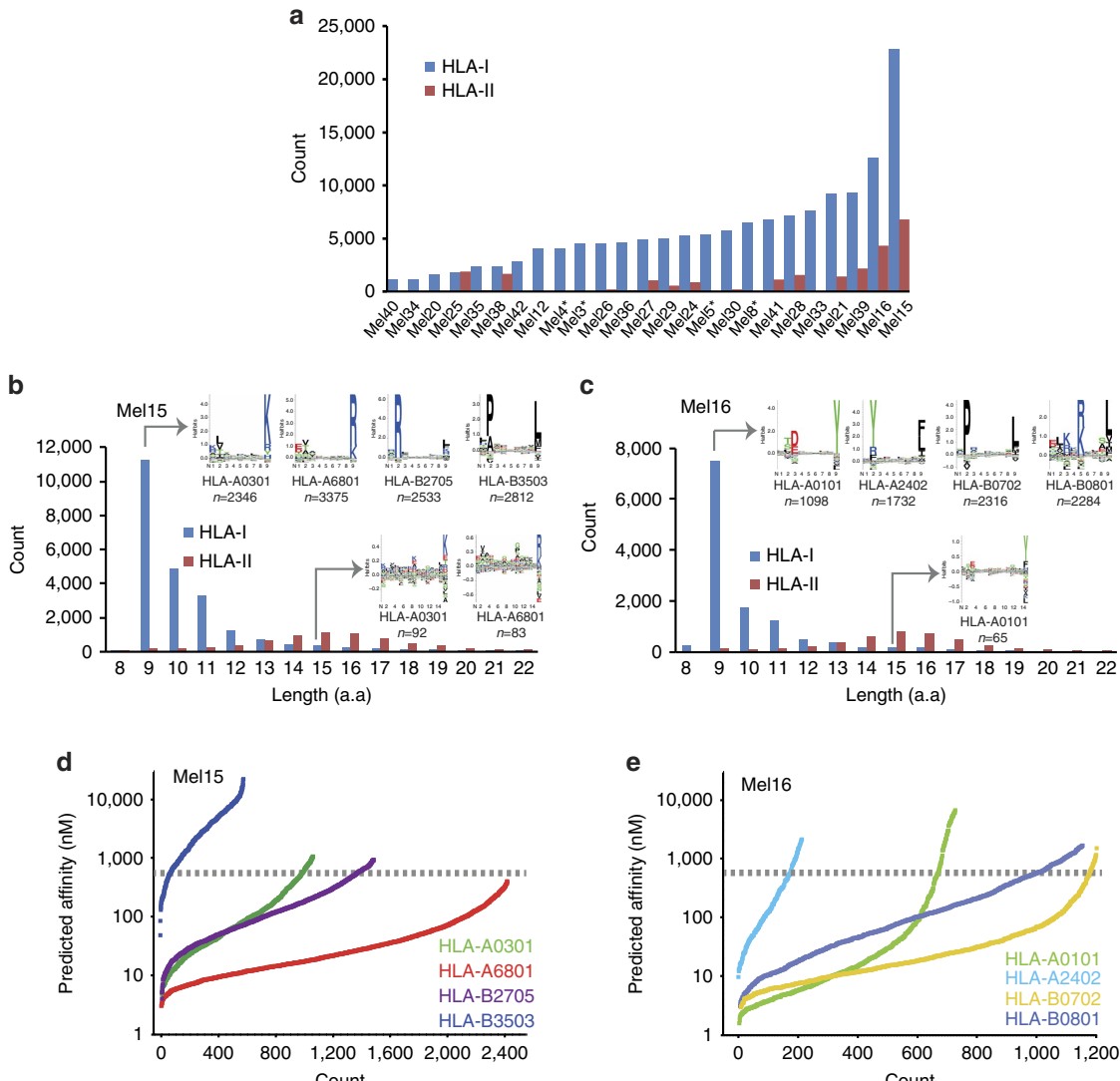

**Figure 1 | In depth analysis of the melanoma-associated ligandome.** Number of epitopes identified per patient. Asterisk marks samples for which no HLA-II peptidomics have been performed (**a**). Typical length distribution of eluted HLA-I and HLA-II peptides in Mel15 and Mel16. HLA-I peptides clustered to reveal the main binding motifs that fit the patients' HLA type (Supplementary Table 2) (**b,c**). Predicted affinity of eluted 9-mer peptides from Mel15 and Mel16 using NetMHC (**d,e**) using the threshold of top 2% ranked predicted sequences. The grey line represents the 500 nM threshold of binding affinity.

detected in both classes I and II peptidome samples (Fig. 2a,b). PMEL-derived peptide ligands were distributed over the entire protein but showed hot spot sequences presented by a large number of patients (Fig. 2b). We normalized for each patient the number of peptides derived from three selected TAA (PMEL, tyrosinase and PRAME) to the total amount of eluted peptides from the respective patient and correlated peptide presentation to RNA and protein expression of the defined TAA (Fig. 2c). These analyses resulted in a statistically significant correlation in case of PMEL (Fig. 2d–g). We observed again that, with respect to peptide prediction, many eluted peptide ligands have predicted binding scores of >500 nM according to NetMHC. Yet, they are still considered among the top 2% (ref. 29) (Supplementary Data 3).

**Phosphorylated peptides are detected without enrichment**. We performed a database search by enabling phosphorylation as a variable modification and although we did not specifically enrich for them, we detected a substantial fraction of phospho-HLA peptides within the eluted immunopeptidome. We filtered the list of identified phospho-HLA peptides by restricting the delta score

to >15 and the localization probabilities to >0.75. After applying such stringent filters, we identified 365 phospho-HLA-I and 25 phospho-HLA-II peptides (Supplementary Data 4). About a quarter of the phospho-HLA binding peptides are shared among at least two patients and 6% of the phospho-HLA peptides have been identified independently in four or more patients' tumour samples (Fig. 3a). One third of the sites have not been previously described in the PhosphoSitePlus database[30] (Fig. 3b). Additional relevant information with respect to these sites is provided in Supplementary Data 4. We observed phosphorylation in 78% of the peptides on Serine and in 19% of the peptides on Threonine. The remaining 3% are on Tyrosine (Fig. 3c). To independently check the accuracy of identification of the phospho-HLA peptides, we synthesized 10 of them and all produced identical MS-fragmentation patterns as compared with the patient derived peptides (Supplementary Figs 2–11). To determine the position of these phosphorylations, HLA-I peptides were grouped according to their length, as presented in Fig. 3d. Interestingly, independent of the broad spectrum of HLA allotypes, the modification is most prominent on the fourth position of 9–11 mer HLA-I peptides and on the fourth and sixth

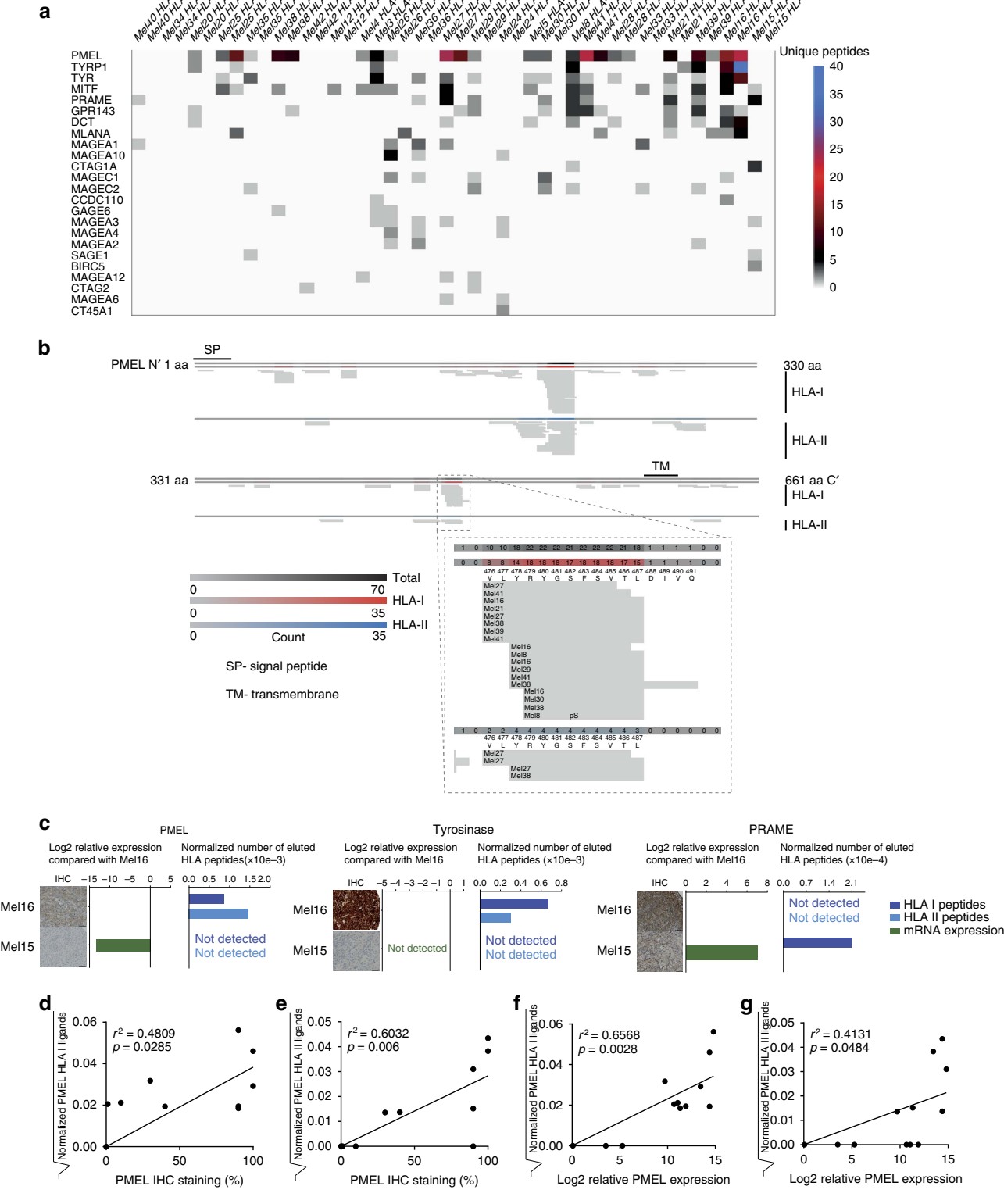

**Figure 2 | Presentation of common tumour-associated antigens.** Heat map presentation of the number of epitopes per patients that derived from a panel of 24 melanoma antigens (**a**). Alignment of the 99 epitopes from PMEL reveals hot spots presented by several patients in common (**b**). Expression of PMEL, tyrosinase and PRAME on mRNA and protein level is exemplarily compared with the number of HLA ligands for patients Mel15 and Mel16 normalized to the total number of identified HLA ligands. mRNA expression is depicted as log2 relative expression as compared with Mel16. Scale bars of Mel16-PMEL, Mel15-Tyrosinase, Mel15- and Mel16-PRAME: 50 µm; scale bars of Mel15-PMEL, Mel16-Tyrosinase: 100 µm (**c**). The number of PMEL-derived HLA class I or II ligands identified by immunopeptidomics was normalized to the total number of identified HLA ligands in the respective patient sample. Square root transformation was applied to deal with deviations from a normal distribution (**c**). Normalized HLA ligand numbers of PMEL of 12 patients with > 2000 HLA I ligands are plotted against per cent positive cells per tissue section as determined by IHC (**d,e**) or against log2 fold mRNA expression relative to a tissue panel consisting of 20 human tissues (**f,g**). Pearson correlation was calculated and the respective p value was corrected for multiple testing. For visual guidance a regression line is depicted on each panel.

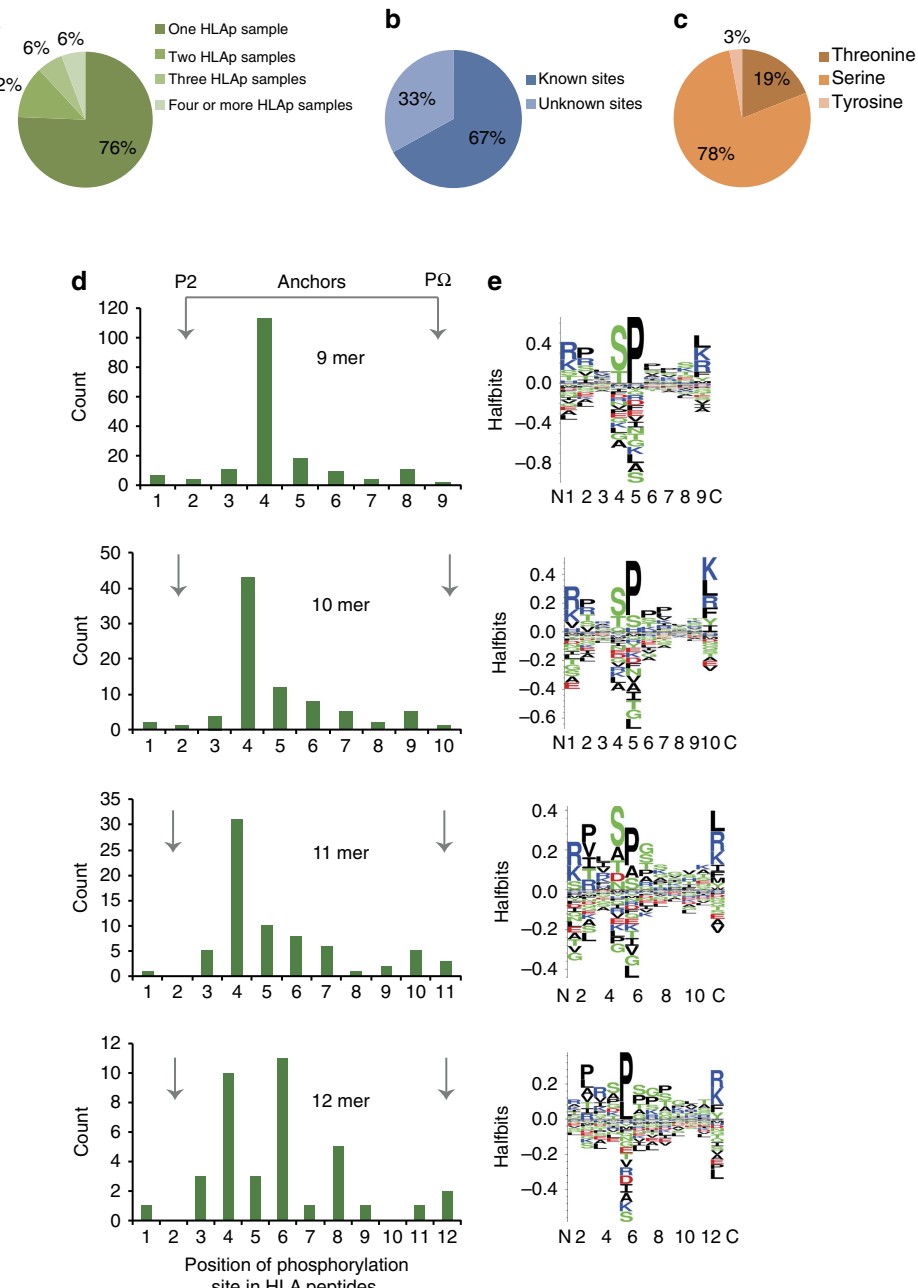

**Figure 3 | Characterization of phosphorylation on eluted HLA peptides.** Percentage of phospho-HLA peptides commonly detected in two or more patients (**a**) as well as percentage of known phosphorylation sites deposited on the PhosphoSitePlus database (**b**). Percentage of defined amino acids affected by phosphorylation within the phosphopeptide ligandome (**c**). Position of phosphorylation within the eluted phospho-HLA peptides according to the peptide length, from 9 mer to 12 mer peptides (**d**). Logo plots of residue frequency at each position of phospho-HLA peptides according to their length (**e**).

positions of 12 mer peptides (Fig. 3d). Moreover, we observed a preferential usage of Arginine and Lysine on position 1 (Fig. 3e) as previously reported for phospho-HLA peptides[31,32]. Of note, a clear signature of proline-directed phosphorylation is apparent in the sequence logos of the phospho-HLA binding peptides (Fig. 3e) as was reported before[32]. These features therefore seem to be rather HLA allotype-independent and make them attractive to be tested as common target antigen candidates in a broader patient population. Taken altogether, our direct approach provides data about a large melanoma-associated phosphopeptide ligandome potentially attractive for targeted immunotherapies.

**Direct identification of mutated peptide ligands by MS**. To test if our method provides the depth to identify peptide ligands possibly comprising mutations, as well as validating them as neoantigens (Fig. 4a), we first performed exome sequencing of the DNA extracted from five patients' tumours exemplarily selected due to variable responses to immune checkpoint modulation. Detailed information about patients and the course of disease is provided in Supplementary Table 2 and Supplementary Fig. 12. Stringent somatic single nucleotide variant (SNV) calling was conducted to define each patient's mutational load and to mimic the state-of-the-art approach for neoepitope prediction (Fig. 4b and Supplementary Data 5). Mutations previously known in

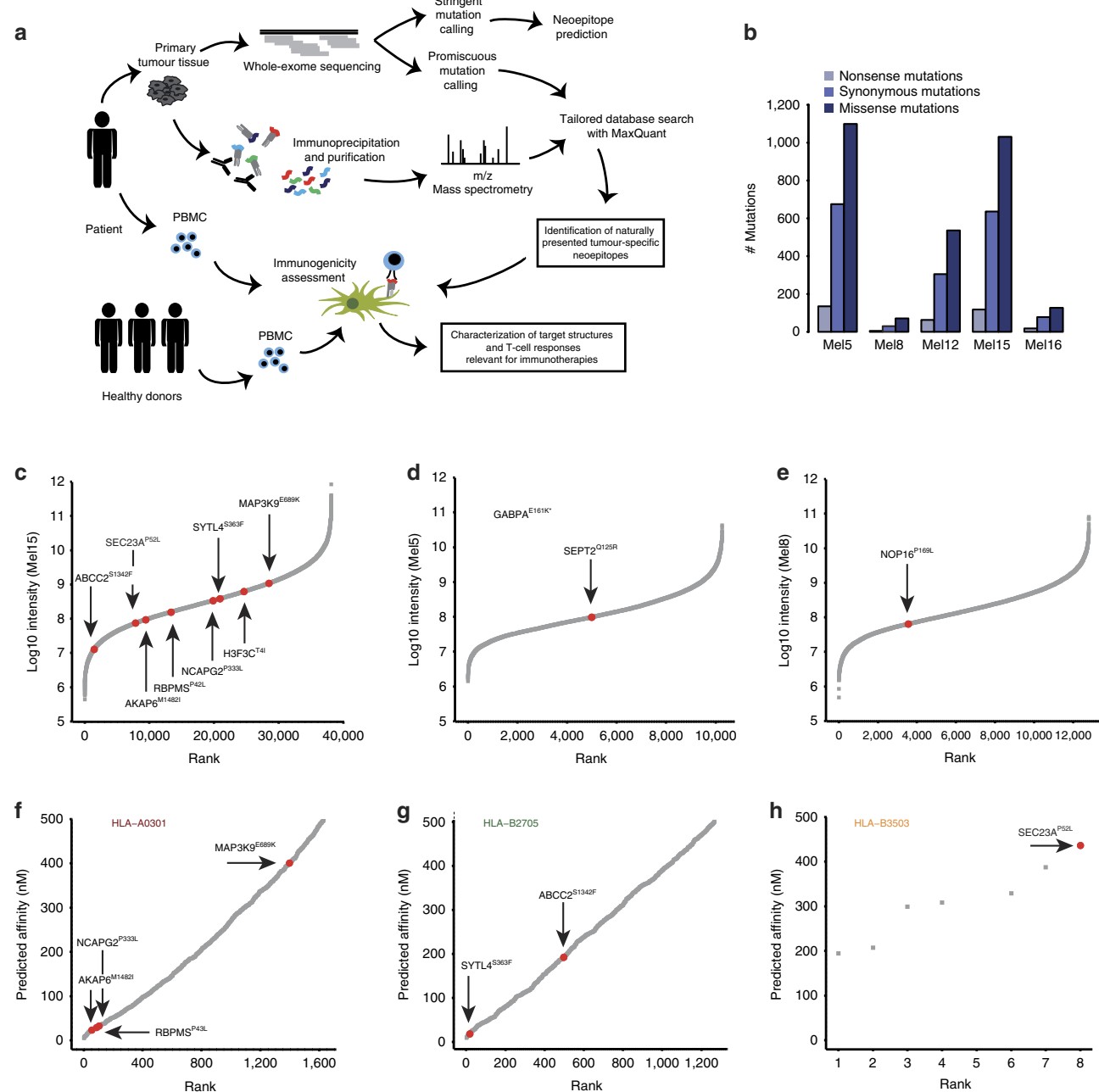

**Figure 4 | Identification of mutated peptide ligands by matching exome sequencing and mass spectrometry immunopeptidomics.** Overview of the experimental approach. Patient tumour tissue was used for MS analysis and exome sequencing. Mutations were called and matched with MS data. Mutated peptide ligands were then further evaluated for recognition by patient's autologous and matched allogeneic T cells (**a**). Overview of the number of non-synonymous and synonymous mutations per patient (**b**). Ranked intensity values of MS data derived from the immunopeptidome of the three patients with identified mutated peptide ligands (Mel15, Mel5 and Mel8). Positions of the mutated peptide ligands are projected on the curve (**c–e**). GABPA$^{E161K}$ was detected at the MSMS level only, therefore no intensity is reported (**d**). Predicted affinity of neoantigen candidates using the 500 nM threshold for binders using NetMHC and ranking of neoepitope candidates for Mel15 with respect to HLA-A0301 ($n = 1,632$), HLA-B2705 ($n = 1,265$) and HLA-B3503 ($n = 8$). The mutated peptide ligands detected by MS are not among the top 10 for HLA-A0301 or – B2705 (**f–h**).

selected tumour probes such as cKIT (P10721.1, L576P; Mel8) and BRAF (P15056.1, V600E; Mel16) were detected by this analysis (Supplementary Data 5). Of note, the mutational load among the five patients differed substantially and neither correlated with the number of identified mutated peptide ligands ($r = 0.65$, 95% CI: −053 to 0.98) nor with the response to immune checkpoint modulation ($r = -0.03$, 95% CI: −0.89 to 0.88) (Fig. 4b, Supplementary Fig. 12 and Supplementary Data 5). In parallel, we developed a new module in the MaxQuant

software that performs mutations calling from NGS data and generates a customized personalized reference database containing all protein isoforms where a detected SNV alters the amino acid sequence. We then performed a non-stringent mutation calling to avoid loss of SNV during database search. This resulted in a high number of non-synonymous mutations in all patients (>15,000 per tumour sample). We searched the raw MS data from the 5 selected patients against this database and, for the first time, directly identified 11 peptide ligands harbouring

**Table 1 | List of 11 mutated peptide ligands identified by MS-based immunopeptidomics from human melanoma tissues.**

| Gene name | Sequence (Position) | a.a Alt | HLA allele predicted affinity nM;%rank;bindLevel | Chr position ENSMBEL transcrip ID | Patient | FDR | Reads tumour Ref:Alt | Reads PBMCRef:Alt | Comments |
|---|---|---|---|---|---|---|---|---|---|
| SYTL4 | GRIAF**F**LKY (358-366) | S363F | HLA-B* 27:05 18.43; 0.6; SB | ChrX:100687163 ENST00000263033 | Mel15 | 1% | 29:9 | 51:1 | WT HLAp GRIAFSLKY detected in Mel15 |
| RBPMS | R**L**FKGYEGSLIK (45-56) | P46L | HLA-A* 03:01 29.2; 0.15; SB | Chr8:30474849 ENST00000517860 | Mel15 | 1% | 63:18 | 122:0 | WT HLAp RPFKGYEGSL; RPFKGYEGSLI; RPFKGYEGSLIKL detected in Mel8 |
| SEC23A | **L**PIQYEPVL (52-60) | P52L | HLA-B* 35:03 436.2; 0.01; SB | Chr14:39095964 ENST00000307712 | Mel15 | 1% | 36:9 | 34:0 | |
| H3F3C | R**I**KQTARK (3-10) | T4I | HLA-A* 03:01 1614; 3.0; -- | Chr12:31792156 ENST00000340398 | Mel15 | 5% | 48:6 | 63:0 | |
| NCAPG2 | K**L**ILWRGLK (332-340) | P333L | HLA-A* 03:01 32.6; 0.15; SB | Chr7:158680743 ENST00000409339 | Mel15 | 1% | 130:23 | 107:1 | |
| AKAP6 | KLKLP**I**IMK (1477-1485) | M1482I | HLA-A* 03:01 23.3; 0.1; SB | Chr14:32822259 ENST00000280979 | Mel15 | 1% | 56:20 | 108:0 | |
| MAP3K9 | ASWVVPIDI**K** (680-689) | E689K | HLA-A* 03:01 400.9; 1.2; WB | Chr14:70733760 ENST00000555993 | Mel15 | 5% | 24:6 | 41:0 | |
| ABCC2 | GRTGAGKS**F**L (1334-1343) | S1342F | HLA-B* 27:05 192.9; 0.7; WB | Chr10:99845661 ENST00000370449 | Mel15 | 5% | 27:10 | 50:0 | |
| NOP16 | SPGPVKLE**L** (161-169) | P169L | HLA-B* 07:02 26.3; 0.12; SB | Chr5:176384171 ENST00000621444 | Mel8 | 5% | 80:11 | 90:0 | |
| GABPA | ETS**K**QVTRW (158-166) | E161K | HLA-A* 25:01 3231.1; 0.40; SB | Chr21:25752162 ENST00000354828 | Mel5 | 5% | 17:22 | 87:0 | WT HLAp ETSEQVTRW detected in Mel5 and Mel40 |
| SEPT2 | YIDE**R**FERY (121-129) | Q125R | HLA-A* 01:01 6.0; 0.01; SB | Chr2:241337414 ENST00000391973 | Mel5 | 5% | 107:77 | 148:0 | WT HLAp YIDEQFERY detected in Mel3, Mel5, Mel8, Mel12, Mel16, Mel25, Mel26, Mel38, Mel39 and Mel40 |

Mutated amino acids within the peptides are indicated with bold letters.

mutations from primary human cancer tissues (Table 1, Supplementary Data 6 and Supplementary Table 3). The mutated peptide ligands have different intensity ranks in the patients' specific tumour ligandomes, and most are within the second and third quartiles of the intensity distribution (Fig. 4c-e). Eight mutated peptide ligands have been identified in the tumour of one single patient (Mel15), a tumour sample for which a large peptidome has been discovered. For this patient, four tumour-derived tissue sections have been processed in parallel for the elution of peptides that afterwards were measured sequentially. Most mutated peptide ligands from patient Mel15 were independently identified in several MS measurements, supporting that they are well detected and well presented in the tumour of this patient. Specifically, SYTL4$^{S363F}$, RBPMS$^{P46L}$, SEC23A$^{P52L}$, MAPK3K9$^{E689K}$ and H3F3C$^{T4I}$ were identified in all four tissue probes, while NCAPG2$^{P333L}$ and AKAP6$^{M1482I}$ were detected in three and two probes, respectively. We synthesized peptides for all HLA ligands representing mutations and found their MS/MS spectra and elution times to be identical to the endogenous ones (Supplementary Fig. 13–24 and Supplementary Table 3). Of note, all of the somatic mutations of the 11 neoepitopes were also detected by the stringent SNV calling. In some cases, we detected the wildtype (wt) peptides, either in the same sample, like wt SYTL4 in Mel15 and wt GABPA and wt SEPT2 in Mel5, or in several other patients' samples, for example GABPA and SEPT2 (Table 1). This might indicate that they are located within hot spots for HLA peptide biogenesis, and since the peptides have been purified from a tissue that contains also healthy cells to a variable degree, presentation of the wt peptides is expected.

**Comparison of identified mutated to predicted peptides.** Prediction of neoepitopes currently represents the standard method to identify mutated peptide ligands potentially representing suitable targets for immunotherapy. To investigate the ranking of our peptides according to standard prediction algorithms, we applied NetMHC (ref. 29) for identification of potential HLA class I-predicted nonamer neoepitopes on all non-synonymous mutations identified by exome sequencing in the tumour probe of patient Mel15. A standard threshold of < 500 nM as predicted affinity was set. We then ranked the predicted affinity and projected the ligandome-based identified mutated peptides on the curve (Fig. 4f-h). Notably, none of the identified mutated peptide ligands were within the top 10 candidates for the HLA allotypes HLA-A0301 (best candidate AKAP6$^{M1482I}$, rank 55) and HLA-B2705 (best candidate SYTL4$^{S363F}$, rank 18) (Fig. 4f-h) for which thorough database information is available. In the case of HLA-B3503 (Fig. 4h), prediction was again highly limited with only eight neoantigens predicted to bind to this HLA allotype.

**Characterization of autologous neoantigen-specific T cells.** We next asked if the MS-detected mutated peptide ligands represent neoepitopes that can be recognized by the patient's own T cells. We selected patient Mel15 as for this patient diverse mutated

peptide ligands were identified and miscellaneous biomaterial could be collected. The detailed clinical course including biomaterial collection of the patient is shown in Fig. 5a. For the investigation of recall responses, we stimulated unfractionated PBMC derived from diverse venipunctures in the course of the disease following application of Ipilimumab (Fig. 5a,b). Without any further enrichment, we identified defined T-cell responses by ELISpot as early as two days after stimulation of PBMC (Fig. 5c). Notably, specific responses were repeatedly observed against SYTL4$^{S363F}$ at that early time point (Fig. 5c). Prolonged peptide stimulation of PBMC derived from diverse blood venipunctures resulted in expansion of T cells with specificity for SYTL4$^{S363F}$, as well as NCAPG2$^{P333L}$ but not for other peptides (Fig. 5d). Dynamic courses of specific responses observed against these two peptides indicate a decline of specific T-cell responses over time. The quality of T-cell responses against SYTL4$^{S363F}$ was superior compared with NCAPG2$^{P333L}$ as shown by higher frequencies of T cells with dual cytokine secretion (Fig. 5e,f). Of note, wt peptides were not recognized (Fig. 5e,f). Specificity of defined T-cell lines was further confirmed by multimer staining of NCAPG2$^{P333L}$–specific T cells (Fig. 5g). In case of T-cell line PBMC-SYTL4-740, we were able to isolate a specific clone, PBMC-SYTL4clone1, which recognized endogenously processed mutated but not wt peptide after minigene transfer of respective gene sequences of SYTL4 (Fig. 5h).

Two years after application of Ipilimumab, a remaining single lung metastasis progressed and was removed at day 796 (Fig. 6a–d). Interestingly, this metastasis showed areas with vital tumour cells with intensive PD-L1 expression while high T-cell infiltration was apparent only in adjacent tumour areas (Fig. 6c,d). PD-L1 may be predominantly responsible for T-cell exclusion and tumour evasion in this case. The defined SYTL4$^{S363F}$ mutation was detected on genomic DNA, as well as reverse transcribed coding DNA (cDNA) level in this second biopsy (Fig. 6e). Importantly, peripheral blood-derived T-cell lines with specificity for SYTL4$^{S363F}$ from day 740 recognized freshly removed tumour material (Fig. 6f). Moreover, ex vivo expanded tumour-infiltrating T cells (TIL) exclusively recognized SYTL4$^{S363F}$ but not other mutated peptide ligands (Fig. 6g). SYTL4$^{S363F}$-specific TIL-derived T-cell responses were functionally sorted and cloned resulting in expansion and further characterization of T-cell clone TIL-SYTL4clone1. Peptide titration of SYTL4$^{S363F}$ revealed a functional avidity in the nanomolar range but no reactivity against the wt peptide (Fig. 6h). Specificity of TIL-SYTL4clone1 was further confirmed by recognition of endogenously processed mutated but not wt peptide as investigated by cytokine release and cytotoxicity (Fig. 6i).

**Validation of neoantigens in the matched allogenic setting**. To investigate if mutated peptide ligands may be immunogenic in matched healthy donors, we stimulated naïve T cells isolated from different donors with mutated peptide ligands. We identified additional reactivity against two peptides, AKAP6$^{M1482I}$ derived from Mel15 and NOP16$^{P169L}$ derived from Mel8 (Fig. 7a). An expanded T-cell line, HD1-AKAP6, with specificity for AKAP6$^{M1482I}$ was further characterized. We observed specific binding of respective multimer but not wt multimer (Fig. 7b). In contrast, peptide titration experiments showed recognition of the mutant but also wt peptide, the latter with reduced functional avidity (Fig. 7c). Functional quality of T-cell responses against wt and mutated peptides were additionally investigated in detail with respect to heterogeneous cytokine release and cytotoxicity (Fig. 7d,e). Therefore, target cells either pulsed with defined peptides or transduced with minigenes were used. Of note,

cytokine responses against wt peptide were inferior when compared with the mutated counterpart whereas the cytotoxic responses were comparable (Fig. 7d,e).

**Discussion**

We hereby present for the first time integrative classes I and II immunopeptidomes of native melanoma tissue samples resulting in the identification of almost 100,000 peptide ligands naturally presented on the tumour. With our methodology >95% of the peptides fit the binding motifs of the different HLA-I allotypes[25], supporting the high yield and purity of the eluted peptides. Also, among the long HLA-I peptides, many seem to fit well to the distinct binding motifs as shown for Mel15 and Mel16. This is in concord to other reports about long HLA-I binders[33–35]. We hypothesize that identical peptides that have been detected in both the class I and II peptidome may be related to common cellular processing which need to be tested in future studies[36,37]. The depth of the ligandome is highlighted by the large number of both, known and novel peptide ligands derived from previously described tumour and melanoma-associated antigens like PMEL, tyrosinase, MELAN-A, NY-ESO-1 and several proteins of the MAGE superfamily of cancer testis antigens. In case of PMEL, from which we detected almost 100 different peptide sequences, the magnitude of presentation, estimated by the number of unique peptide ligands per peptidome sample correlated with messenger RNA (mRNA) and protein expression. Alignment of the PMEL derived HLA peptides on the PMEL protein sequence revealed that several domains along the protein sequence are sources of multiple class I and II peptides in several tumour samples derived from diverse patients. We collectively name such domains as 'hot spots'. Other domains may not be as efficiently accessible for the antigen processing and presentation machinery and those were either not presented at all by any of the 25 studied melanoma tumours, or their resulting peptides were below our detection limit. Targeting of PMEL by peptide-based vaccination showed only limited clinical success when compared with results of checkpoint modulation[4] and combination of anti-CTLA-4 treatment with PMEL vaccination did not enhance anti-tumour activity[9]. We hypothesized that our large dataset might shed light on the extent these peptides are presented *in vivo*. Interestingly, the two nonameric peptides, P209 and P280, used previously for vaccination were eluted only from tumour probes of Mel27. In the case of P209, the nonamer peptide sequence is indeed located in the most dominant hot spot for presentation, although the sequence was, with exception of Mel27, included in peptide ligands with a length >14 aa. In the case of P280, the peptide sequence could be detected only in Mel27. These data suggest other PMEL-derived peptides to be potentially more promising for defined targeting in a larger patient cohort.

Even without further enrichment, peptide ligands harbouring posttranslational modifications as phosphorylation were detected. This implies for the high recovery and sensitivity of our method and importantly it avoids the requirement of reservation of dedicated samples for enrichments of phospho-peptides and additional laborious sample processing[38]. Nevertheless, such peptides may contain cancer-specific phosphorylation patterns and therefore potentially represent attractive targets for cancer immunotherapy[31,32]. One third of identified phosphorylation sites have not been reported in the PhosphoSitePlus database[30]. We envision that direct immunopeptidomic analyses have the potential to identify novel sites on protein sequences that may not be compatible with trypsin digestion and therefore may be undetected by shotgun phospho-proteomics[39]. 24% of the phospho-HLA peptides have been identified in tumour samples

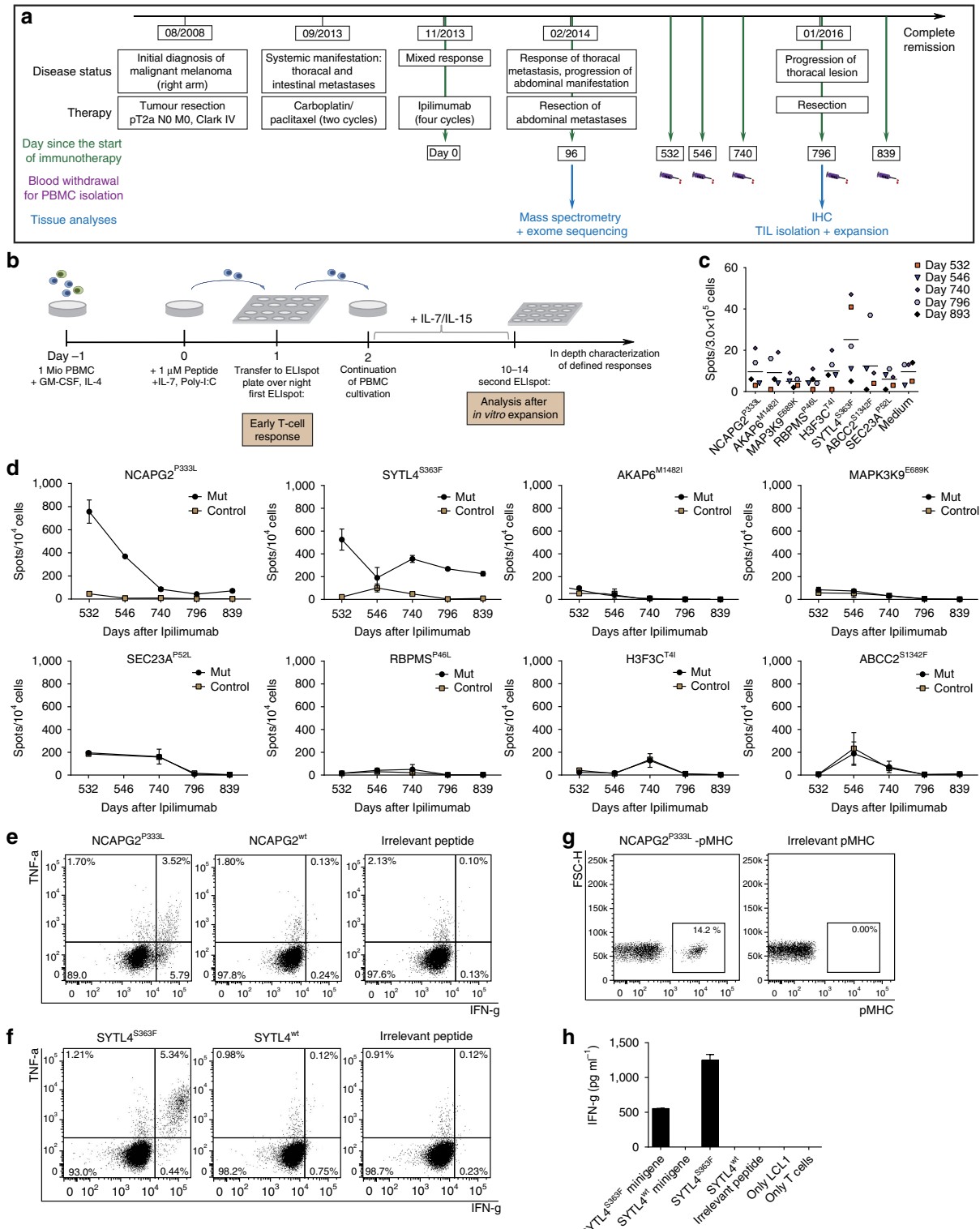

**Figure 5 | Immune responses against mutated ligands in PBMC of patient Mel15.** Clinical course and retrieval of patient material (**a**). Schematic overview of the experimental design of recall immune responses among blood-derived T cells from patient Mel15 (**b**). Early immune responses detected in PBMC derived from different blood withdrawals two days after *in-vitro* peptide stimulation (**c**). Time course of specific reactivities of blood-derived PBMC obtained at different time points against the eight identified mutated epitopes from patient Mel15. All analyses were performed in duplicates and spot counts were adjusted to $10^4$ cells (**d**). Intracellular cytokine staining (ICS) of an expanded NCAPG2[P333L] specific T-cell line from day 546 (PBMC-NCAPG2-546) after co-incubation with peptide pulsed T2-A3 target cells for 5 h (**e**). ICS of T-cell line PBMC-SYTL4-740 stimulated with SYTL4[S363F] from day 740 after co-culture with peptide pulsed T2-B27 target cells (**f**). Staining of line PBMC-NCAPG2-546 with the specific multimer in comparison to irrelevant multimer staining (**g**) IFN-g secretion after coincubation of T-cell clone PBMC-SYTL4clone1 derived from line PBMC-SYTL4-740 with peptide pulsed and minigene-transduced LCL1 (results of triplicates) (**h**). Data from experiments with triplicates are shown as mean ± s.d., data resulting from duplicates are shown as mean.

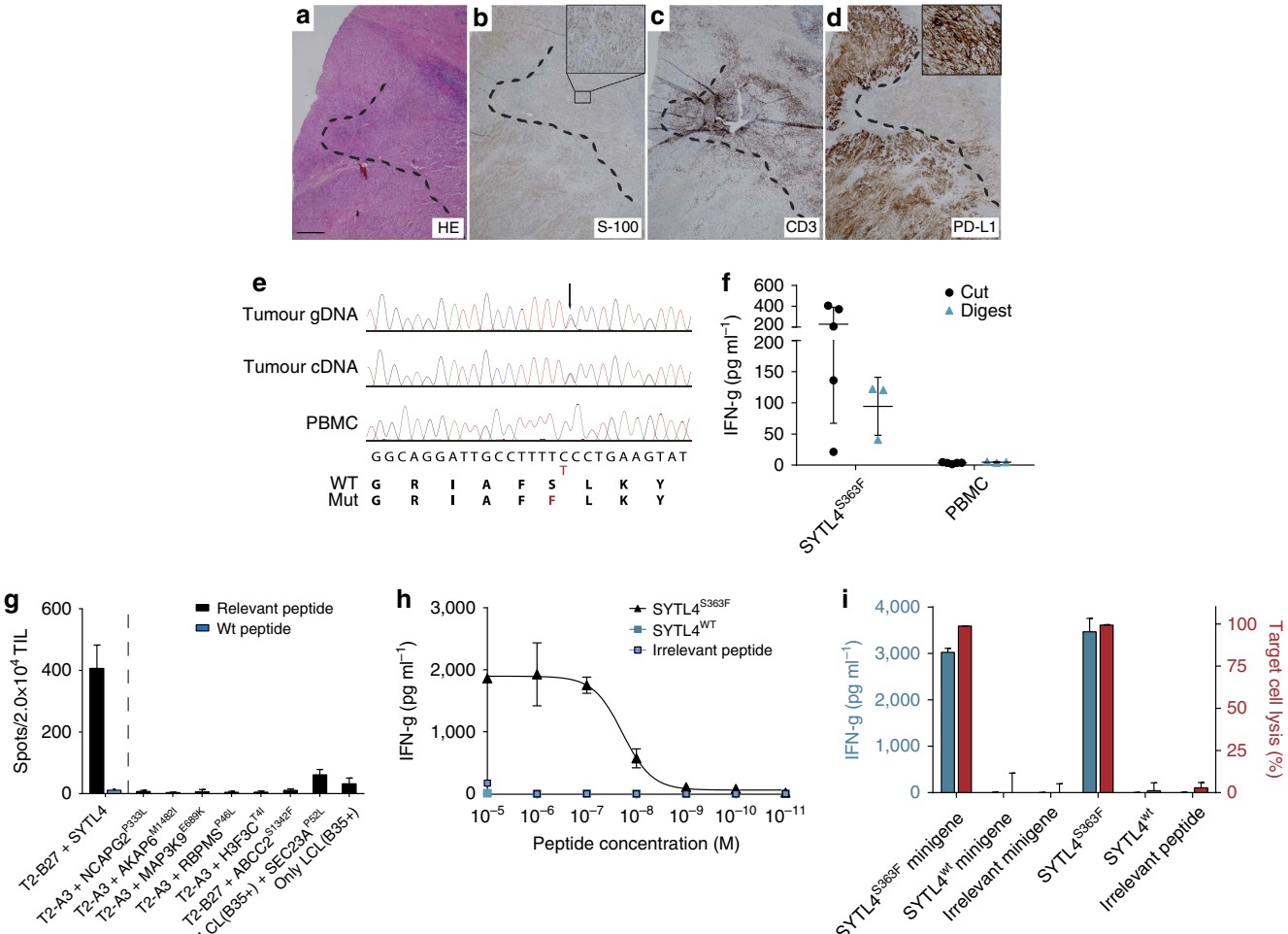

**Figure 6 | In-depth characterization of tumour and peptide-reactivity of SYTL4-specific T cells derived from PBMC as well as TILs.** HE (**a**) staining of a lung metastasis after metastasectomy (01/2016, day 796) as well as immunohistochemistry stainings with anti-S100 (**b**), anti-CD3 (**c**) and anti-PD-L1 (**d**); (**b,d**) Inset: ×20 magnification. Scale bar, 500 μm. Sanger sequencing of the mutated region from SYTL4 in processed tumour material from day 796 using either isolated genomic DNA (gDNA) or coding DNA (cDNA) as template (**e**). IFN-g secretion of T-cell line PBMC-SYTL4-740 on co-culture with cut (5 wells) or digested (3 wells) fresh tumour material for 36 h (**f**). Non-stimulated PBMC from Mel15 served as controls. Horizontal lines and error bars show mean and s.d., respectively. Co-incubation of *in-vitro* expanded TIL with target cells pulsed with mutated peptide ligands (**g**). SYTL4wt served as negative control, analysis was performed using triplicates and depicted as mean ± s.d. Reactivity of the TIL-derived T-cell clone TIL-SYTL4clone1 against T2-B27 target cells pulsed with titrated concentrations of mutated, wt or irrelevant peptide (**h**). Co-culture of TIL-SYTL4clone1 with LCL1 either peptide pulsed or transduced with mutated or wt minigenes (**i**) with results shown as mean of duplicates. Amount of IFN-g secretion was assessed in supernatants (left *Y*-axis) and amount of target cell lysis was analysed by FACS-based acquisition of total number of vital target cells in relation to untreated LCL (right *Y*-axis); coincubation was performed in triplicates and results are shown as mean and s.d.

derived from multiple patients. A clear signature of proline-directed phosphorylation of the detected HLA peptides could be observed and this is likely to be assigned to a defined kinase motif associated to cell proliferation and tumorigenicity[32,40]. Thus, our data point to a common oncogenic phospho-peptide signature potentially attractive for multimodal targeting. Notably, our data revealed a very conserved motif within detected phospho-HLA peptides with preferred Arginine and Lysine in P1 and the phosphorylation site in P4. This canonical motif has been previously described for defined HLA allotypes and structural data suggest that the conserved amino acid usage in P1 may increase peptide binding of low affinity peptides whereas phosphorylation in P4 may improve immunogenicity by direct presentation of the phosphorylation site to the TCR or conformational peptide changes[31]. Our data indicate that such peptides are presented over a broad HLA repertoire making these peptides attractive to be tested as more general target

antigens. Reactivity of patients' derived autologous T cells with specificity for these peptides might be limited due to negative thymic depletion of reactive T cells. However, TCR derived from the mismatched or xenogeneic repertoire may still represent attractive therapeutic tools to target self-antigens with cancer-specific expression in adults[41].

In contrast to shared TAA, mutated peptide ligands can be regarded as foreign antigens which have been previously described to be well detectable by autologous T cells in diverse disease settings[16,17,19,20,42–44]. Especially clonogenic ones have been shown to be associated to a durable clinical benefit by immune checkpoint inhibitors[24]. We hereby describe for the first time the identification of mutated peptide ligands derived from patients' non-modified and non-cultivated native tumour tissue samples using our discovery MS approach. Within this proof of concept, with testing of five patients, we detected 11 mutated peptide ligands, 8 of them in one patient. This

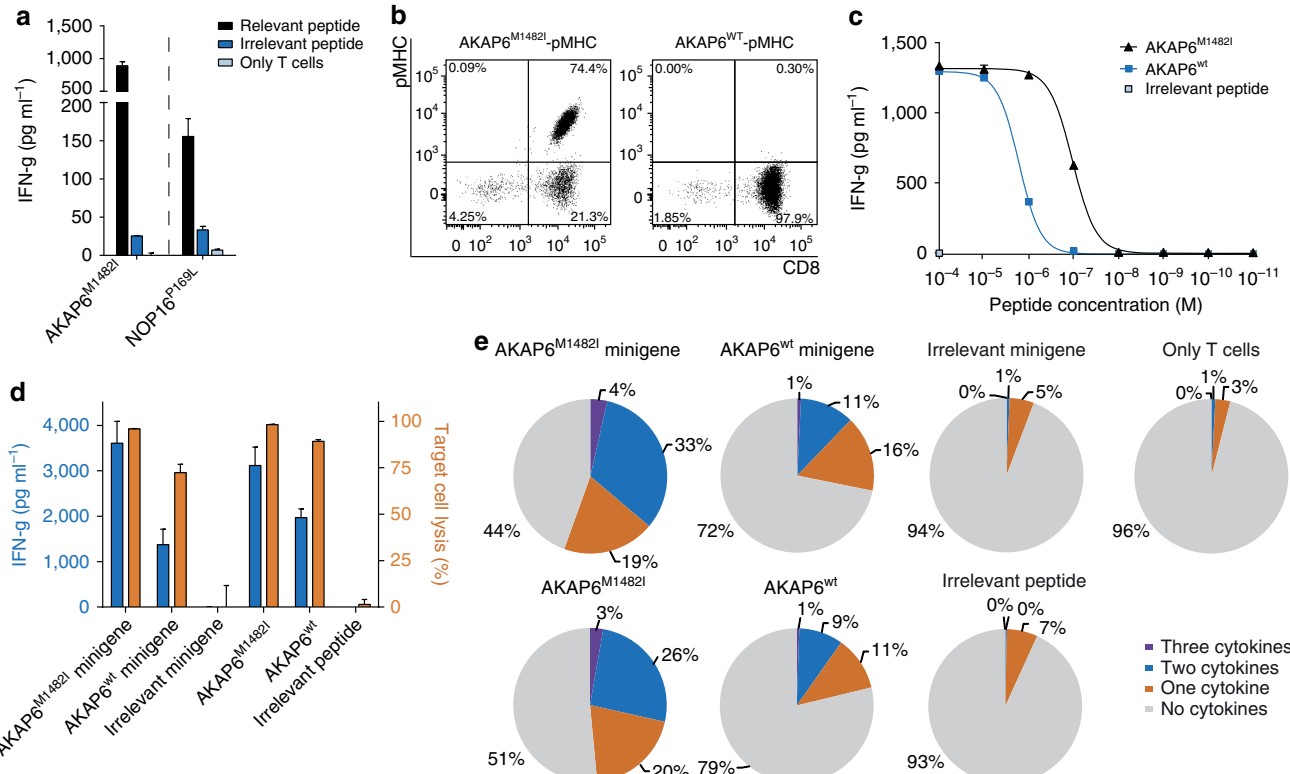

**Figure 7 | Characterization of mutant-specific T-cell responses in HLA-matched healthy donors.** T-cell responses of two different matched healthy donors against neoepitopes AKAP6[M1482I] and NOP16[P169L]. Effector cells were coincubated in duplicates with T2-A3 or T2-B7 pulsed either with the relevant peptide or control peptides with the same HLA restriction as the mutated ligands, results are shown as mean (**a**). Staining of T-cell line HD1-AKAP6 with the mutated or wt multimer (**b**). IFN-g release of the T-cell line on peptide titration of AKAP6[M1482I] and its non-mutated counterpart using T2-A3 as targets (duplicates are depicted as mean) (**c**). IFN-g secretion (left Y-axis) and target-cell lysis (right Y-axis) after coincubation of the T-cell line HD1-AKAP6 with peptide-pulsed and minigene-transduced LCL1 cells performed in triplicates, data shown as mean ± s.d. (**d**). Intracellular cytokine staining (IFN-g, TNF-a and IL-2) on co-culture of the T-cell line HD1-AKAP6 with LCL1 cells, either peptide-pulsed or minigene-transduced, determined by flow cytometry. Cells were gated on ethidium monoazide bromide-negative and CD8-positive events (**e**).

patient experienced prolonged clinical benefit following the application of Ipilimumab, a checkpoint inhibitor involved in the enhancement of primary and memory T-cell responses[45]. Notably, response to treatment in melanoma has been associated to the mutational load, suggesting that mutated peptide ligands are the major source of target antigens[13]. Few mutated peptide ligands have been previously identified by MS using murine or human cell lines as raw material[21–23]. However, the direct identification of neoantigens from non-modified native human tumour tissue represents an important breakthrough for several reasons. Tissue, unlike cell lines, is heterogeneous. Although more challenging, mutated peptide ligands identified from this material are likely among the most well presented peptides and hence the best targets for immune interventions. In fact, most mutated peptide ligands identified from Mel15 were independently identified in several MS measurements of different tissue probes, and their measured MS intensity indicates adequate presentation similar to non-mutated self-antigens. MS-based detection is by nature biased to detect the more abundant peptides and hence will favour identification of clonogenic mutated peptides. In contrast, mutations present only in malignant subclones are likely to be under-represented in the total peptidome and therefore below the detection limit of the current MS-based discovery approach.

Most of the neoantigens we detected are predicted to bind with high affinity to their respective HLA molecules, although they are mostly not within the top 10 predicted ones. In case that amino acid alterations are located in anchor positions, novel ligands may be generated. However, in case that the alterations are not in anchor positions there is a fair chance that also the wt peptides will be detected by MS, especially if the peptides are expected to bind with high affinity. Indeed, we detected the corresponding wt peptides of mutated SYTL4 and GABPA in the peptidomes of Mel15 and Mel5, respectively. Interestingly, we also detected corresponding wt peptides, and also sequences that are shorter or longer versions of the core neoantigen sequences in HLA peptidome samples of other patients with alternative HLA allotypes as e.g. for RPBMS. Multiple peptide sequences homing to a certain location on the protein suggest that this might be again a hot spot for presentation. Thus, mutations that are included in hot spots may have preferred presentation in vivo, although larger studies are needed to confirm this hypothesis. However, if this hypothesis is correct, future large-scale immunopeptidomics studies are expected to reveal such hot spots in the human proteome, and consequently in-silico algorithms should prioritize neoantigen candidates that are included within them in order to shorten the target list.

The potential and promise of MS detection of neoantigens is in fact highlighted by the hit rate of mutated peptide ligands with obvious clinical relevance. In fact, two out of eight mutated peptide ligands were detected by blood and TIL-derived autologous T cells of patient Mel15. This indicates a clear advantage compared with the usage of prediction software tools to identify neoantigens[16,19,24,44,46,47]. Moreover, mutation-

specific T-cell responses could be detected as early as two days after peptide stimulation without prior enrichment as previously published[44,46]. Mutation-specific T-cell lines recognized freshly isolated tumour material further emphasizing clinical relevance of these neoantigens. Of note, neoantigen-specific responses were declining over time whereas a single remaining lung metastasis started to progress. These data are highly intriguing and suggest that mutation-specific T-cell responses might be investigated as personalized surrogate biomarkers in future studies. Two other mutated peptide ligands were recognized by matched allogeneic T cells and we suggest that such matched allogeneic T cells may be an attractive source to be used for adoptive T-cell or TCR transfer as recently published and suggested[43]. Interestingly, in-depth characterization of one defined T-cell population demonstrated reactivity also against the wt peptide. The distinct T-cell population was not detected by the wt multimer and wt-peptide-specific cytokine release was clearly inferior if compared with the mutated peptide. However, lysis of minigene-transduced or peptide-pulsed target cells presenting respective wt epitope reached high levels comparable to lysis of targets presenting the mutated ligand. These data suggest that differences in TCR avidity in response to mutated versus wt peptide may follow similar rules compared with observed differences in response to viral antigens[48]. Thus, multi-functional characterization of neoantigen-specific T cells is important to estimate risks of autoimmunity and neoantigen-specific matched allogeneic T-cell populations or TCR need to be carefully selected for adoptive transfer.

The time necessary for direct identification of mutated peptide ligands by MS using native tissue probes can be as short as three weeks, including whole exome sequencing analysis. Therefore, this approach is highly suitable for the development of personalized treatment approaches. In contrast, the usage of cell lines as raw material for MS analysis requires prolonged time for generation of a sufficient number of cells and is not always successful. Similarly, the prediction approach currently predominantly used for identification of presumable neoantigens is highly time consuming and expensive as subsequent large-scale immunogenicity testings are necessary for neoantigen validation. Moreover, the prediction approach harbours the risk for biased or limited results especially in case of binders to rare HLA allotypes. Our data indicate that the commonly used threshold for predicted affinity of 500 nM is by far too low for some HLA allotypes. Moreover, our data may be highly useful as to be a training dataset to further improve the performance of such predictors and consequently to enable more reliable in-silico identification of neoepitopes in the future.

The issue of the sensitivity of this discovery MS-based approach is currently the major limitation as indicated by the limited number of identified neoantigens. More than that, we could not detect neoantigens among the class II peptidome in this study. The latter result was expected as class II molecules are typically expressed on professional antigen presenting cells in the tumour microenvironment, and often not directly on the melanoma cells. Currently, unlike T-cell based assays, the MS approach is not sensitive enough to detect the few copies of neoantigens presented only on professional antigen presenting cells that are in the tumour microenvironment. A more intensive fractionation of the HLA peptides sample prior to the MS analysis may increase the depth but will also significantly increase the investment in MS measurement time. We envision that the new generations of MS instruments, new computational algorithms and more efficient procedures for sample preparations will further improve the sensitivity and therefore this direct antigen discovery approach.

The direct identification of clinically relevant antigens, both shared and private, will foster our understanding of essential characteristics of targets and their respective specific T cells relevant for effective tumour rejection and protection. Defined neoepitopes can be targeted by vaccination and respective T-cell responses can be tracked and used as biomarkers. Neoepitopes and defined common TAA not recognized by the patient's T cells may be attractive for alternative immunotherapeutic strategies such as transfer of effector cells with defined specificity.

## Methods

**Primary human material and cell lines.** Informed consent of all healthy and diseased participants was obtained following requirements of the institutional review board (Ethics Commission, Faculty of Medicine, TU München). All patients included in the analysis were diagnosed for metastatic malignant melanoma and treated at the Klinikum rechts der Isar, TU München. An overview about all patients is given in Supplementary Table 1. More detailed information is provided for patients Mel5, Mel8, Mel12, Mel15 and Mel16 who additionally donated blood for isolation of PBMC and identification of mutated peptide ligands by matching the immunopeptidome with exome sequencing data (Supplementary Fig. 12, Supplementary Table 2 and Supplementary Data 5). Tumour tissue samples were collected from patients, who underwent tumour resection at the Department of Surgery, Klinikum rechts der Isar of the TU München. Immediately after resection (within 30 min), tumour tissue was macroscopically dissected by an experienced pathologist, snap frozen and stored in liquid nitrogen ($-196\,^\circ$C) at the MRI-TUM-Biobank (MTBIO) until usage. Additional tumour tissue was formalin-fixed and paraffin-embedded (FFPE). Before molecular analysis, tumour diagnosis was confirmed by a pathologist and tumour content was determined by an HE stain taken from the sample going to be used. TIL derived from the tumour tissue of patient Mel15 removed at day 796 after treatment with Ipilimumab were expanded for 2–3 weeks by cultivation of minced tumour tissue pieces with irradiated feeder PBMC, 1,000 U ml$^{-1}$ IL-2 (PeproTech, London, UK) and 30 ng ml$^{-1}$ OKT3 (kindly provided by Elisabeth Kremmer). Change of medium supplemented with 300 U ml$^{-1}$ IL-2 was performed twice a week. PBMC from patients and healthy donors were isolated from whole blood by density-gradient centrifugation (Ficoll/Hypaque, Biochrom, Berlin, Germany) immediately on receipt. T cells were cultivated in T-cell medium, RPMI 1640 (Invitrogen, Carlsbad, CA, USA) supplemented with Penicillin/Streptomycin (Pen/Strep) (PAA, Pasching, Austria), 5% FCS, 5% human serum, 10 mM Hepes (Invitrogen) and Gentamycin (Biochrom, Berlin, Germany), or serum-free AIM-V (Invitrogen) as indicated. Cell lines used in this study: T2 (American Type Culture Collection (ATCC), Manassas, VA), lymphoblastoid cell lines (LCL) LCL1 (IHW09005, HLA-A0301, B2705, C0102) and LCL2 (IHW09216, HLA-A0201, A0301, B3502, B3801 (both kindly provided by Steve Marsh). Morphology and constant growth behaviour of all cell lines were controlled periodically and the absence of mycoplasma infection was routinely confirmed by PCR (Venor GeM mycoplasma detection kit, Minerva Biolabs). T2 were retrovirally transduced with the HLA restriction elements HLA-A0301 (T2-A3), B0702 (T2-B7) or B2705 (T2-B27) as described before[49]. All target cell lines were maintained in RPMI 1640 supplemented with Pen/Strep and 10% FCS.

**Purification of HLA peptides.** For the preparation of the affinity columns, panHLA-I and panHLA-II antibodies were purified from HB95 cells and HB145 cells (ATCC, Manassas, VA), respectively. We cross-linked the antibodies to Protein-A Sepharose beads (Invitrogen, CA) with 20 mM dimethyl pimelimidate in 0.2 M sodium borate buffer pH9. Tumour amount that has been available for this research varied significantly, from about 0.1 g to $4 \times 1$ g in Mel15. For the purification of HLA complexes, snap-frozen melanoma tissue samples were homogenized for 10 s on ice using ULTRA-TURRAX (IKA, Staufen, Germany) in a tube containing 5–10 ml of lysis buffer and incubated at 4 $^\circ$C for 1 h. The lysis buffer contained 0.25% sodium deoxycholate, 0.2 mM iodoacetamide, 1 mM EDTA, 1:200 Protease Inhibitors Cocktail (Sigma-Aldrich, MO), 1 mM PMSF, 1% octyl-β-D glucopyranoside (Sigma-Aldrich, MO) in PBS. The lysates were cleared by 20 min centrifugation at 40,000g. Lysates were passed through a column containing Protein-A Sepharose beads (Invitrogen, CA) to deplete the endogenous antibodies. Subsequently, HLA-I molecules were immunoaffinity purified from cleared lysate with the W6/32 antibody covalently bound to Protein-A Sepharose beads (Invitrogen, Camarillo, CA). HLA-II molecules were then purified by transferring the flow through onto similar affinity columns containing the HB-145 antibody. Affinity columns were washed first with 10 column volumes of 150 mM NaCl, 20 mM Tris–HCl (buffer A), 10 column volumes of 400 mM NaCl, 20 mM Tris–HCl, 10 volumes of buffer A again, and finally with seven column volumes of 20 mM Tris–HCl, pH 8.0. HLA molecules were eluted at room temperature by adding 500 µl of 0.1 N acetic acid, in total seven elutions for each sample[25].

Eluted HLA peptides and the subunits of the HLA complexes were loaded on Sep-Pak tC18 (Waters, MA) cartridges that were prewashed with 80% acetonitrile (ACN) in 0.1% trifluoracetic acid (TFA) and with 0.1% TFA. The peptides were separated from the much more hydrophobic HLA heavy chains and B2M on the

C18 cartridges by eluting them with 30% CAN in 0.1% TFA. They were further purified using a Silica C-18 column tips (Harvard Apparatus, Holliston MA) and eluted again with 30% ACN in 0.1% TFA. The peptides were concentrated and the volume was reduced to 15 µl using vacuum centrifugation. Remaining immunoaffinity purified HLA heavy chains and the B2M molecules were eluted from the Sep-Pak tC18 cartridges with 80% ACN in 0.1%TFA. For western-blot detection, 1% of each of those protein containing samples were used. Anti human B2M antibody EP2978Y (1:5,000, Abcam, Cambridge, United Kingdom) was used and was detected with donkey anti-rabbit IgG HRP conjugate secondary antibody (1:5,000, Thermo Fisher Scientific) in a peroxidase assay using SuperSignal West Pico Chemiluminescent substrate (Thermo Fisher Scientific).

**LC–MS/MS analysis of HLA peptides.** HLA peptides were separated by a nanoflow HPLC (Proxeon Biosystems, Thermo Fisher Scientific, Odense) and coupled on-line to a Q Exactive or the Q Exactive HF mass spectrometers (Thermo Fisher Scientific, Bremen) with a nanoelectrospray ion source (Proxeon Biosystems). We packed a 50 cm long, 75 µm inner diameter column with ReproSil-Pur C18-AQ 1.9 µm resin (Dr. Maisch GmbH, Ammerbuch-Entringen, Germany) in 100% methanol. Peptides were eluted with a linear gradient of 2–30% buffer B (80% ACN and 0.5% acetic acid) at a flow rate of 250 nl min$^{-1}$ over 90 min. Data were acquired using a data-dependent 'top 10' method, which isolated them and fragment them by higher energy collisional dissociation. We acquired full scan MS spectra at a resolution of 70,000 at 200 m/z with a target value of 3e6 ions. The most intense ions were sequentially isolated and accumulated to an AGC target value of 1e5 with a maximum injection time of 120 ms. For measurement of HLA-I peptides, in case of unassigned precursor ion charge states, or charge states of four and above, no fragmentation was performed. In addition, we excluded the fragmentation charge state of one from measurement of HLA-II peptides. The peptide match option was disabled. MS/MS resolution was 17,500 at 200 m/z. Fragmented m/z values were dynamically excluded from further selection for 20 s.

**Synthetic peptides.** Synthetic peptides for spectra validation were synthesized with the Fmoc solid phase method using the ResPepMicroScale instrument (Intavis AG Bioanalytical instruments, Cologne, Germany).

**Mass spectrometry data analysis of HLA peptides.** We employed the MaxQuant computational proteomics platform[25,26] version 1.5.3.2. Andromeda, a probabilistic search engine incorporated in the MaxQuant framework[50], was used to search the peak lists against the UniProt databases (Human 85,919 entries, Sep 2014), and a file containing 247 frequently observed contaminants such as human keratins, bovine serum proteins, and proteases. For identification of mutated peptide ligands, customized references databases were used (see below). N-terminal acetylation (42.010565 Da), methionine oxidation (15.994915 Da) and phosphorylation (79.9663304 Da on serine, threonine and tyrosine) were set as variable modifications. The second peptide identification option in Andromeda was enabled. The enzyme specificity was set as unspecific. Andromeda reports the posterior error probability and FDR, which were used for statistical evaluation. A false discovery rate of 0.01 was required for peptides for the global ligandome analysis and for the phospho-HLA peptides identification. We applied in addition a less stringent threshold of 5% for the identification of mutated peptide ligands. No protein false discovery rate nor permutation rules were set in MaxQuant in creating the decoy database. The initial allowed mass deviation of the precursor ion was set to 6 p.p.m. and the maximum fragment mass deviation was set to 20 p.p.m. We enabled the 'match between runs' option, which allows matching of identifications across different replicates that belongs the same patient, in a time window of 0.5 min and an initial alignment time window of 20 min. From the 'peptide.txt' output file produced by MaxQuant, hits to the reverse database and contaminants were eliminated. The resulting list of peptides is provided in Supplementary Data 2.

**Variant search using MaxQuant.** For the analysis of peptides we used a newly designed module of the MaxQuant software that enables the search for peptides based on genomic variations. MaxQuant takes as input aligned reads from exome data and calls variants as described below. The base search space results from unspecific digestion of all protein sequences utilizing all peptides from length 8 to 25. Variants increase the peptide search space by either including or excluding them on each peptide. In case several variants can be present on the same peptide all combinations of the absence/presence patterns are taken into account. In extreme cases of very many combinatorial possibilities for a peptide, these are cut off at 100 contributing peptides. To account for different a priori probabilities of different peptide classes the posterior error probability is calculated depending on the type of the peptide. For instance, different classes of peptides that are treated separately in the posterior error probability calculation are unmodified peptides without variants, unmodified peptides resulting from a variant, phosphorylated peptides without variants and phosphorylated peptides resulting from a variant. A common PSM-FDR threshold is applied based on this peptide class dependent posterior error probability.

**DNA isolation from FFPE tissue.** For isolation of genomic DNA (gDNA) from FFPE tissue, paraffin sections (five 10 µm sections per tumour sample) were de-paraffinized using xylene (2 × 5 min) and cleared in absolute ethanol. Tumour tissue was macro-dissected and DNA isolation was performed with DNeasy Blood & Tissue Kit (Qiagen/Hilden, Germany) according to manufacturer's instructions with following modifications: (i) tissue lysis was performed for an extended period of 60 h and (ii) Qiagen MinElute spin columns were used for a reduced elution volume of 50 µL.

**Whole-exome sequencing and bioinformatics analysis.** DNA fragmentation was performed with Covaris S2/E220 ultrasonicator to yield a fragment size of ~200 bp. The SureSelectXT Human All Exon V5 (Agilent Technologies, Santa Clara, USA) kit was used for library preparation. Sequencing (100 bp paired-end) was performed on the Illumina HiSeq 2000 system. Mutation calling was performed according to a promiscuous or stringent protocol. For promiscuous mutation calling, we excluded positions with quality <13 (equivalent to 0.05 error probability) and used the following thresholds: total read depth of the position should be >10 reads; number of reads which support a variant should be greater than 5 reads and at the same time the minimum variants frequency was set to 5 per cent.
Stringent variant calling was done with Mutect v1.1.7 (ref. 51) using default settings. Mutations were considered as relevant if the frequency was greater or equal 5% and the read depth was greater or equal 10. Raw read sequences were filtered with Prinseq v0.20.4 (ref. 52). Nucleotides with a Phred Score below 20 at 3′ or 5′ end were clipped. Reads were then mapped to the GRCh38.p3 (http://www.ensembl.org/index.html) reference genome with BWA v0.7.12 (ref. 53) using default settings. Duplicates were marked with Picardtools v1.129 (http://picard.sourceforge.net.) and kept for downstream analysis. Realignment and base recalibration was done with GATK v3.3 (ref. 54). Annotation was done with SNPeff Version 4.1g (ref. 55) based on the ENSEMBL GRCh38.78 genome. Only transcripts with CCDS sequences were used for further analysis.

**Semi-quantitative realtime PCR.** Tumour tissue from melanoma patients was micro-dissected and RNA extraction was performed according to the manufacturer's instructions using High Pure RNA Paraffin Kit (Roche Diagnostics/Mannheim, Germany). As control, RNA from the Human Total RNA Master Panel II (Clontech, Mountain View, USA), from human testis (Clontech) or human adult skin (amsbio, Abingdon, U.K.) was used. cDNA was synthesized by the Superscript II reverse transcriptase (Invitrogen) using random hexamer primers (Roche). qPCR was conducted in a StepOnePlus system (Applied Biosystems) using the KAPA Probe Fast Universal qPCR MasterMix (peqlab, Erlangen, Germany). Relative quantification was calculated by the delta-delta Ct method[56] using the geometric mean of control genes (GAPDH, HMBS and HPRT1) for normalization. The following primers and probes were used: PMEL: 5′-ACCTATCCCTGAGCCTGA AG-3′ (forward primer (fwd)), 5′-GCCCAGGGAACCTGTAATACT-3′ (reverse primer (rev)), 5′-[6FAM]TGCCAGCTCAATCATGTCTACGGA[TAM]-3′ (probe); Tyrosinase: 5′-TGCACAGATGAGTACATGGGA-3′ (fwd), 5′-GGCTAC AGACAATCTGCCAAG-3′ (rev), 5′-[6FAM]CTCAGCCCAGCATCATTCTT CTCCT[TAM]-3′ (probe); PRAME: 5′-TATCGCCCAGTTCACCTCTC-3′ (fwd), 5′-ATCACGTGCCTGAGCAACT-3′ (rev), 5′-[6FAM]CAGTCTGCAGTGCCTGC AGGC[TAM]-3′ (probe); GAPDH: 5′-TTCCAATATGATTCCACCCA-3′ (fwd), 5′-GATCTCGCTCCTGGAAGATG-3′ (rev), 5′-[6FAM]TTCCATGGCACCGTC AAGGC[TAM]-3′ (probe); HMBS: 5′-ACGATCCCGAGACTCTGCTTC-3′ (fwd), 5′-GCACGGCTACTGGCACACT-3′ (rev), 5′-[6FAM]CCTGAGGCACCTGGA AGGAGGCTG[TAM]-3′ (probe); HPRT1: 5′-CTGGCGTCGTGATTAGTGAT-3′ (fwd), 5′-CTCGAGCAAGACGTTCAGTC-3′ (rev), 5′-[6FAM]CATTATGCTG AGGATTTGGAAAGGGTG[TAM]-3′ (probe).

**Immunohistochemistry.** FFPE tumour samples were selected to construct a tissue microarray using a Tissue Microarrayer (Beecher Instruments/Sun Prairie, USA) with a core size of 0.6 mm. At least three tumour cores from tumour center and tumour periphery were taken from areas previously marked by a pathologist.
Immunohistochemistry was performed on 2 µm sections using the following antibodies: S-100 (polyclonal, dilution 1:600, DAKO, Hamburg, Germany), HMB45 (clone HMB-45, dilution 1:200, Cell Marque, Rocklin, USA), MelanA (clone A103, dilution 1:200, Cell Marque, Rocklin, USA), PRAME (polyclonal, dilution 1:150, Sigma-Aldrich), Tyrosinase (clone T311, dilution 1:200, Santa Cruz, Dallas, Texas). Immunohistochemistry on one representative slide of the pulmonary metastasis was performed using the following antibodies: S-100 (Polyclonal, dilution 1:600, DAKO, Hamburg, Germany), CD3 (Clone MRQ 39, dilution 1:500, Cell Marque, Rocklin, USA) and PD-L1 (Clone 28-8, dilution 1:500, Abcam, Cambridge, United Kingdom). Stainings were run on an automated immunostainer with an iVIEW DAB detection kit (Ventana Medical Systems, Roche Diagnostics, Mannheim, Germany). Appropriate positive controls for each antibody were run in parallel. In cases of marked staining heterogeneity, 2 µm sections from FFPE tumour blocks were stained in addition to exclude scoring inaccuracy due to tumour heterogeneity. Immunoreactivity was evaluated regarding the percentage of positive tumour cells. Nuclear and cytoplasmic staining was taken into account. A 4-tiered system was used for scoring: (0) absent, (1) >0–25%, (2) >25–50%, (3) >50–75%, (4) >75–100%.

**Statistics.** For the analysis of correlation of ligand identification and antigen expression, the square root of the normalized number of PMEL HLA ligands was calculated to deal with deviations from a normal distribution. Pearson correlation was calculated and the respective $p$ value was corrected for multiple testing. A regression line is depicted for visual guidance on each panel.

**HLA typing.** HLA typing was done for selected patients on gDNA isolated from PBMC by next generation sequencing (Zentrum für Humangenetik und Laboratoriumsdiagnostik, Martinsried, Germany) or using the HLA miner tool[57] for exome sequencing data when limited patient material was available.

**Sanger sequencing of DNA and RNA from tumour samples of Mel15.** Snap-frozen tumour tissue obtained from the resection at day 792 was homogenized by mechanical disruption. Genomic DNA was obtained using DNA mini kit (Qiagen). RNA was extracted by passing sheared tissue additionally through a QIAshredder Homogenizer (Qiagen) followed by isolation with RNeasy mini kit (Qiagen). Reverse transcription was performed with Affinity Script (Agilent) and oligo(dT) Primers. PCR was conducted with KOD Polymerase (Merck Millipore) and Primers as described for minigene cloning (see below). Products were purified after gel electrophoresis with Nucleospin Gel and PCR Cleanup kit (Macherey-Nagel) and sequenced at MWG Eurofins (Ebersberg, Germany).

**Algorithms used for prediction of peptide ligands.** Affinity to the corresponding allotypes expressed in Mel15 was predicted for all eluted peptides identified in Mel15 samples using NetMHC4.0 (ref. 29). To be more conservative regarding assignment of peptides with multiple specificities, the list of peptides was filtered to include only 9 mer peptides that bind to only one HLA allotype. The threshold for binding was set to rank $<2\%$ to include weak binders (standard setting according to ref. 58). This resulted in 1,065, 2,518, 1,499 and 581 peptides that fit HLA-A0301, HLA-A6801, HLA-B2705 and HLA-B3505, respectively. Predicted affinities to the HLA supertype representative allotypes were calculated for the TAA-derived peptides using NetMHCcons[59], and are provided in Supplementary Data 3. Clustering of peptides into groups based on sequence similarities was performed using the GibbsCluster-1.1 tool[27] using default settings.

For prediction of affinity scores of mutated peptide ligands, protein transcript sequences associated with non-synonymous mutations were downloaded from ENSEMBL GRCh38.78. A 23-mer small peptide sequence was generated by adding 11 amino acids up and downstream of the altered position. If the mutation is located less than 11 amino acids away from the 3′ or 5′ end, the peptide is shorter, respectively. The peptide was then used as input for NetMHC4.0 (ref. 29) and fragments comprising the mutation were used for further analysis. Ligands with a predicted affinity of $<500$ nM were included in the graphical analysis.

**Cloning and expression of minigenes.** Oligonucleotide primers were designed to amplify fragments of gene products ranging between 200 and 400 bp encompassing the mutated base. Generally, forward primer additionally encoded a methionine and the reverse primer contained a stop codon to allow generation of a mutated and non-mutated version of the respective minigene for immunological assays. gDNA isolated out of FFPE melanoma tissue was used as template for PCR amplification of the minigene containing the defined mutation whereas PBMC served for cloning of the wt minigene, except for the NCAPG2-derived mutated and wt minigenes, which were cloned out of customized synthesized vector constructs (GenScript, Piscataway, USA). Following primer sequences were used for cloning of all constructs: AKAP6_fwd 5′- TAGCGGCCGCCACCATGGATGAG GGGGAAAGCAT-3′, AKAP6_rev 5′- TAGTCGACTTCTATATTGCCACTTTT AT-3′; NOP16_fwd 5′- TAGCGGCCGCCACCATGTCCTCGATTCTTTGCA G-3′, NOP16_rev 5′- TAGTCGACAGAGCGGGGAGTGTGCACGT-3′ (control minigenes); SYTL4_fwd 5′- TAGCGGCCGCCACCATGAGTACGATCGGCAGC AT-3′, SYTL4_rev 5′- TAGTCGACCTTGGCTTCATCAGCATAGG-3′; NCAPG2_fwd 5′- TAGCGGCCGCCACCATGTCTCCAGTGCATTCCAA-3′, NCAPG2_rev 5′- TAGTCGACCATGAAGGTTTGGATCC-3′ (control minigenes). Cells were transduced with retroviral vectors coding for the respective minigenes and the fluorescent dye dsRed Express II to allow sorting of transgenic cells. Retroviral particles were generated as described previously[60]. Briefly, retroviral vector plasmids coding for respective minigenes were co-transfected with plasmids carrying retroviral genes for gag/pol derived from Moloney murine leukaemia virus (pcDNA3.1-Mo-MLV) and env (pALF-10A1) into 293T cells using TransIT (Mirus, Göttingen, Germany). After 48 h incubation, supernatants were filtered (45 μm) and used for transduction of LCL1.

**In-vitro stimulation of effector T cells.** Recall antigen-experienced T-cell responses were investigated by stimulation of PBMC from patients or healthy donors as previously described[61] with slight modifications. Briefly, 0.3–0.5 Mio PBMC per well were cultivated in AIM-V for 24 h in the presence of IL-4 and GM-GSF (PeproTech, London, UK). 1 μM Peptide (GenScript, Piscataway, USA), 0.5 ng ml$^{-1}$ IL-7 (Peprotech) and 20 μg ml$^{-1}$ Poly-I:C (Invitrogen) were added after 24 h. Cells were then transferred to a previously coated IFN-g ELISpot plate

and cultured over night at 37 °C. Afterwards, cells were gently re-suspended and re-cultivated in T-cell medium.

For stimulation of naive T cells with defined mutated peptide ligands, monocytes of healthy donors were differentiated into dendritic cells (DC) by plate adherence and incubation with IL-4 (20 ng ml$^{-1}$) and GM-CSF (100 ng ml$^{-1}$) (Peprotech) for 48 h. Cells were further matured using TNF-a (10 ng ml$^{-1}$), IL-1b (10 ng ml$^{-1}$), IFN-g (5,000 IU ml$^{-1}$), PGE$_2$ (250 ng ml$^{-1}$) (Peprotech) and CL075 (1 μg ml$^{-1}$) (InvivoGen, San Diego, USA) for 24 h. Naïve T cells from the DC donor were isolated as described previously[49]. After pulsing of DC with 1 μM peptide for 2 h in AIM-V medium (Invitrogen), priming was started at an effector to target ratio of 10:1 in the presence of IL-21 (30 ng ml$^{-1}$) (Peprotech). In each stimulation procedure, IL-7 and IL-15 (5 ng ml$^{-1}$) (Peprotech) were added every two to three days.

**Multimer staining and further enrichment of specific T cells.** HLA multimers were manufactured as previously described[62]. Multimer staining was performed according to current recommendations and protocols of the CIMT Immunoguiding Program (http://www.cimt.eu/workgroups/cip).

For a detailed investigation of T-cell responses against SYTL4 on the clonal level (Figs 5h and 6h,i), expanded T-cell lines were functionally sorted by enrichment of CD137 positive cells after overnight stimulation with irradiated peptide pulsed T2 cells, cloned by limiting dilution and screened for peptide-specific recognition. Relevant clones (PBMC-SYTL4clone1, TIL-SYTL4clone1) were further expanded using irradiated feeder, IL-2 and Okt-3 for assessment of minigene recognition and peptide titration assays.

**Functional T-cell analysis.** Expanded T cells were co-incubated after 10–14 days with peptide-pulsed (1 μM) target cells or cell lines transduced with different minigene constructs. Respective target cells were pulsed with the mutated peptides, the wt counterpart or irrelevant peptides with the same HLA restriction as ligands of interest (designated as control peptides). Coincubation assays for detection of cytokine secretion were performed in duplicates. ELISpot analysis was performed with IFN-g-coating monoclonal antibody (1-D1K), IFN-g-capture-mAb (7-B6-1-biotin) and Streptavidin-HRP (all Mabtech, Sweden) as recommended by the manufacturer using 20,000 target cells and 20,000–40,000 effector cells per well as indicated. Phorbol 12-myistate 13-acetate (PMA) (Sigma-Aldrich) and Ionomycin (Merck, Germany) were used for a positive control. ELISpot plates were read out on an ImmunoSpot S6 Ultra-V Analyzer using Immunospot software 5.4.0.1 (CTL-Europe, Bonn, Germany).

Co-culture experiments for assessment of IFN-g release were performed with an effecter-to-target ratio of 1:1 using each 10,000 target and effector cells per well. Peptide titration assays were performed at least twice showing comparable results for each reactivity pattern. IFN-g release in cell culture supernatants of coincubation assays was determined using the BD OptEIA Human IFN-g ELISA Kit II (BD Biosciences, Franklin Lakes, USA). Intracellular cytokine staining was performed with IC staining kit (eBioscience). 100,000 effector cells were coincubated with 100,000 target cells. After one hour, 10 μg ml$^{-1}$ Brefeldin A (Sigma-Aldrich) was added and cells were incubated for 4 more hours at 37 °C. Cells were then stained with Ethidium-monoazide bromide (Invitrogen) for life-dead discrimination and anti-CD8-APC (clone RPA-T8). After fixation and permeabilization, intracellular cytokines were stained with anti-IFN-g-AF700 (clone B27), anti-TNF-a-V450 (clone Mab11) and anti-IL-2-BV510 (clone 5344,111) antibodies (all BD Biosciences).

Cytotoxic activity of specific T cells was analysed by coincubation of 50,000 effector cells with 50,000 target cells followed by using FACS-based quantification of remaining target cells after 20 h. Therefore, cocultures of target and effector cells were stained with 7-Aminoactinomycin D (7AAD) (Sigma-Aldrich) for dead cell exclusion and anti-CD8-FITC (clone HIT8a) and anti-CD3-AF700 (clone UCHT1) (all BD Biosciences). Target cells were identified according to their morphology in the FSC/SSC and gated on 7-AAD$^-$/CD8$^-$/CD3$^-$ events. Absolute numbers of cells per well were calculated with AccuCheck COUNTING BEADS (Invitrogen) according to the manufacturer's instructions. Lysis of minigene transduced or peptide-pulsed LCL was then set in relation to untreated LCL cocultered with respective effector cells using the following formula:

$$\text{percentage of lysed LCL} = \left(1 - \frac{\text{absolute number of remaining LCL}}{\text{mean of untreated LCL}}\right) * 100$$

(1)

Cytotoxic experiments were performed in triplicates and depicted results are representative for two independent experiments each. Measurements of all FACS-based assays were performed on a LSR II flow cytometer (BD Biosciences) and samples were analysed using FlowJo Software.

T-cell responses against freshly removed tumour material from patient Mel15 were analysed by using either small non-treated tumour pieces or digested tumour tissue. Non-treated fresh material was prepared by mincing tumour tissue into small pieces of 1 mm length and two tumour fragments were added to each well of a 96-well plate. Tumour digestion was performed as described previously[63] with slight modifications. Briefly, teased tissue ($<3$ mm$^3$) was incubated with tumour digestion medium consisting of RPMI supplemented with DNase type I, Hyaluronidase, Collagenase type IV (all Sigma-Aldrich), Pen/Strep and

Gentamycin. Obtained tumour suspension was cocultured with 50,000 cells of expanded T-cell lines or 100,000 cells of freshly isolated PBMC per well.

**Data availability.** The mass spectrometry proteomics data have been deposited to the ProteomeXchange Consortium via the PRIDE partner repository[64] with the dataset identifier PXD004894. Whole exome sequencing data has been deposited at the European Genome-phenome Archive (EGA), which is hosted by the EBI and the CRG, under accession number EGAS00001002050. The authors declare that all the other data supporting the finding of this study are available within the article and its supplementary information files and from the corresponding author on reasonable request.

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

## Acknowledgements

This work was supported by grants from the Wilhelm Sander-Stiftung (2015.030.1), the DFG/SFB824 (C10) and DFG TR/SFB36 (A13). M.B.-S. was supported by the Alexander von Humboldt-Foundation. We are highly thankful to our patients for their cooperation. We also thank Stephanie Rämisch for excellent technical support and Chloe Chong for experimental support during revision. There are no competing interests for patenting.

## Author contributions

M.B.-S., R.K., M.M., A.M.K. designed the study, M.B.-S., E.B., R.K., S.A., J.W., M.S., J.S.-H., K.S. and A.M.K. performed and/or analysed experiments, D.H.B. generated MHC multimer reagents, T.E., P.S., R.R. and J.C. performed statistics and bioinformatics, R.H., A.W., M.E.M., C.P. and A.M.K. provided patient material, M.B.-S., E.B., R.K., T.E., R.R., M.M. and A.M.K. wrote the manuscript.

## Additional information

**Competing financial interests:** The authors declare no competing financial interests.

**Publisher's note**: 

