## [Peer Review File · Nature Communications]

Comments:

1. minor edits for English grammar need to be made
2. There is a spectrum of peptide counts detected across the tumors (Fig. 1a). How do we know that the entire dataset for each tumor represents saturation and not simply technical variation? The authors should provide a measure to indicate level of saturation given peptide diversity.
3. Fig 1 legend. Please label the grey threshold line....What does this represent?
4. tetramer assays should be used to show that the neoantigen peptide expanded T cells are specific.

Reviewer #2 : Expert in Antigen proteomic
(Remarks to the Author):

A. Summary of the key results

The manuscript by Bassani-Sternberg et al. explores the immunopeptidomes of melanoma biopsies and provides a rich data set from which to examine the presence of tumour specific peptides, although it should be noted that healthy tissue was not included in the analysis and therefore the absolute tumour specificity is imputed from exome sequencing. In addition to finding peptides from known tumour antigens such as CT-antigens, phosphopeptides that may represent dysregulated signalling in the cancer cells and mutated peptides associated with tumour specific mutations which could play a role in tumorigenesis were identified.

B. Originality and interest: if not novel, please give references

There is no doubt this is a hot area of research and although overplayed in the text this is one of the first studies to identify mutated peptides/neoepitopes. In addition the immunogenicity of such peptides was also examined.

C. Data & methodology: validity of approach, quality of data, quality of presentation

The approach is excellent and data quality is high as one might expect from such high profile laboratories.

D. Appropriate use of statistics and treatment of uncertainties

This is OK

E. Conclusions: robustness, validity, reliability

Overall the conclusions of this manuscript are valid, although the discussion is poorly written and difficult to follow in places. A thorough edit of the text is required to aid clarity and improve grammatical errors.

The statement (p16 lns 289-292) that immunopeptidomics studies sample trypsin sensitive sites is not substantiated and should be removed or altered to clearly indicate this is speculation

Some details regarding the patient therapy and background would help understand some arguments. The anti-CTLA-4 therapy facilitates activation of T cells presumably through natural priming from tumour derived antigens. This is not indicated and for those not in the immunooncology field this argument should be clearly defined and appropriate back ground given. This is the same issue with the discussion of PMEL immunised patients in previous studies, not enough background to allow the reader to reach the same conclusion as the authors.

F. Suggested improvements: experiments, data for possible revision

1) Edit of text to improve readability

2) Details of the amount of tumour biopsy used for the peptide elutions should be included and more details regarding its preparation prior to IP provided.

3) What do the authors think the peptides that bind to both class I and class II represent? Have they formally shown these are not simply contaminants or copurifying class I/classII IP'd with the reciprocal mAbs? Binding studies showing class II and class I binding would be reassuring otherwise I am not convinced this is a justifiable category.

- 4) The failure to be able to predict accurately the affinity of peptides bound to HLA allomorphs with little training data is a valid point but it begs the question is Fig 1d and e necessary?
 - 5) In Fig 2d it is unclear what is being examined and how the peptide abundance was normalised? Or does this represent the number of ligands from a given antigen? If the latter what about peptide abundance?
 - 6) P.11 In 213 - it is not clear what hotspot refers to?
 - 7) A more thorough explanation of the T cell assays in the text is required. It is hard to follow what data represents direct ex vivo analysis using ELISpot and what represents in vitro expanded T cell specificities. Please provide a more in depth narrative.
 - 8) P17 In 329 - which self-antigens are you referring to?
- G. References: appropriate credit to previous work?
n/a
- H. Clarity and context: lucidity of abstract/summary, appropriateness of abstract, introduction and conclusions
See above

Reviewer #3 : Expert in melanoma immunotherapy
(Remarks to the Author):

This is a report of a technological improvement of cancer associated antigen-derived peptide identification using tandem mass spectrometry. The experiments are meaningful and well-controlled.

The ability to identify peptides encoded by mutated genes is important and may help the development of novel cancer immunotherapies and biomarkers. However, the current study focuses on methodological advancement, without applying the new ability to identify mutated and phosphopeptides, limiting its overall impact. In essence, this is a methodological advancement. The detection of NCAPG2 and SYTL4-specific T cells, for example, is heartening, but it is not clear that these T cells were involved in the patient's disease course. Indeed, the authors show generation of neoepitope-specific T cells from normal donors; it is now widely accepted that T cells against many self and foreign peptides can be generated from normal donors and cancer patients alike. Similarly, while the authors may be correct that the prevalence of phosphorylation on position 4 of nonapeptides, may mean that these phosphopeptides are attractive targets for immunotherapy across HLA types (line 304), is a speculation, without any evidence in the present manuscript. Finally, the analysis of T cell responses to neoepitopes in 1 patient in Fig. 5 does not appear to add much beyond what has been previously published on T cell recognition of neoepitopes by other groups.

Overall, the increased sensitivity of the peptide identification approach by Bassano-Sternberg et al. is an important advancement, but the lack of application of the method to bring forward fundamentally new understanding or application limits the impact of the current study.

Point-by-point-reply

We would like to first thank the reviewers for their helpful and constructive comments and appreciation.

Reviewer #1 : Expert in Neo-antigens in cancer
(Remarks to the Author):

Review of Bassani-Sternberg et al.

In this manuscript, the authors examine neoepitopes from native human melanoma tissue using mass spectrometry. Neoantigens are important functional targets for immunotherapy, including immune checkpoint inhibitors and neoantigen based vaccines. However, identification of presented neopeptides has proven challenging so far. In silico prediction is only partially accurate and improvement in mass spec based identification of presented neopeptides is needed. In this paper, the authors present deep mass spec data for 25 melanoma tumors for MHC found neoepitopes. This paper is a very nice study providing data for the field on the immunopeptidome as well as technical advances for MHC mass spec detection of neoantigen peptides. This is a useful study for the field.

Comments:

1. minor edits for English grammar need to be made
We revised the text accordingly.
2. There is a spectrum of peptide counts detected across the tumors

(Fig. 1a). How do we know that the entire dataset for each tumor represents saturation and not simply technical variation? The authors should provide a measure to indicate level of saturation given peptide diversity.

We used for all IPs experiments exactly the same amount of beads and antibodies. As this amount was sufficient to purify equivalent of 20,000 peptides in Mel15 (1%FDR), this setup is far from being saturated for all the remaining melanoma samples in this study. The number of peptides eluted from each tumor is certainly dependent on diverse factors. Tumor cell content, HLA expression and defects in antigen expression may influence the richness of the repertoire of HLAp. Technical aspects related to detection by MS are also involved to some extent, for example the fact that peptides with charged amino acids tend to be detected more efficiently and some HLA molecules have a preference to bind such peptides and hence better overall representation in the peptidome. However, the most important factor is the amount of tissue that was used for the extraction of the peptides. The variability observed in Fig. 1a is mainly related to the sample amount, as tissue amount varied from about 0.1 g to 4x1 g (in case of Mel15). As also from very low tissue material there is a chance to detect tumor associated peptides, we decided to include and report all 25 patients in this study. We performed additional Western Blot to correlate recovery of beta-2 microglobulin (B2M) from the original affinity purifications to demonstrate the amount of recovered HLA complexes. We show that the amount of B2M is significantly and positively correlated to the total number of HLA-I peptides in this study. We have now included these data and information in the result and method section.

3. Fig 1 legend. Please label the grey threshold line....What does this represent?

We have labeled the line accordingly indicating a threshold affinity of 500nM.

4. tetramer assays should be used to show that the neoantigen peptide expanded T cells are specific.

We agree with the reviewer that detected T-cell responses need to be carefully validated. Therefore, we have performed multiple assays for defining and confirming specificity of the isolated and expanded T-cell populations dependent on available T-cell material and reagents (see Table below). Novel data have been now included in the revised manuscript investigating the specificity of T-cell responses more in detail. All reactivities were confirmed by at least one alternative experimental setting.

	MHC restriction	ELISpot	ELISA	Intracellular cytokine secretion	Multi-mer staining	Reactivity against minigenes	Tumor reactivity	Cyto-toxicity	T-cell clones
SYTL4 (S363F)	B2705	+	+	+	n.a.	+	+	+	+
NCAPG2 (P333L)	A0301	+	+	+	+	n.d.	(+)	n.d.	-
AKAP6 (M1482I)	A0301	+	+	n.d.	+	+	n.d.	+	+
NOP16 (P169L)	B0702	+	+	n.d.	n.a.	n.d.	n.d.	n.d.	-

n.d. not determined; n.a. not available; (+) only minor, statistically non-significant reactivity has been observed (data not shown in the manuscript); green: autologous T-cell responses observed in patient Mel15; red: T-cell responses from matched healthy donors

Autologous neoantigen-specific T-cell responses against SYTL4^{S363F} and NCAPG2^{P333L}:

Screening of circulating antigen-specific T cells was performed by ELISpot assays offering a highly sensitive method for detection of specific T cells. ELISpot analyses were performed as early as 2 days after peptide stimulation of non-selected PBMC resulting in detectable T-cell responses against two mutated peptide ligands of the tumor of Mel15. Peptide specificity for the two epitopes, SYTL4^{S363F} and NCAPG2^{P333L}, was confirmed 10 days after peptide-specific T-cell expansion again by ELISpot as well as subsequently by ELISA assays using expanded T-cell lines and clones. In case of NCAPG2^{P333L}, specificity was now further confirmed by multimer analysis. In case of SYTL4^{S363F}, multimer production was unfortunately so far not successful. However, diverse T-cell populations with defined specificity could be isolated from the peripheral blood as well as TILs demonstrating peptide specificity as well as recognition of endogenously processed peptide after minigene transfer as investigated by multifunctional assays. These novel data have been now included in Fig. 5 and Fig. 6.

Donor-derived matched T-cell responses against mutated peptides:

Donors with defined matched HLA alleles were selected for stimulation assays performed after isolation of donor-derived naive T cells. Isolation of defined T-cell populations from a naive T-cell repertoire requires a selective choice of suitable donors with matching HLA alleles and a time-consuming stimulation procedure. Out of 36 stimulation procedures we isolated T cells recognizing two mutated peptide ligands. We characterized reactivity against one epitope (AKAP6^{M1482I}) more in detail and provided more information in the novel version (Fig. 7). As shown before, we have observed cytokine secretion also in response to the wildtype (wt) peptide although to a lower extent. This has been also demonstrated by a lower functional avidity of the T-cell line in response to peptide-

pulsed target cells. Interestingly, the wt multimer does not bind to the selected T-cell population. In contrast, we observed similar cytotoxic responses against wt and mutated peptide. These data indicate that (i) diverse assays will be necessary to properly characterize reactivity against mutated and wt peptide similar to virus-specific responses (7) and (ii) that lower avidity T-cell responses against wt peptides lacking full effector functions may still harbor a risk for cross- and autoreactivity.

In case of NOP16^{P169L}, we have currently no more T-cell material for additional tests due to the limited growth of defined T-cell lines.

We certainly will enhance our efforts in generating multimers as well as isolating T cells with defined specificities. Although we are convinced that we may be able to provide more data also including additional multimer analyses in a reasonable time frame, we do not think that these data would add much to the current information at this point. We have already a highly comprehensive and detailed characterization of defined T-cell responses and also show limitations of multimer assays with respect to AKAP6^{M1482I}-specific T-cell populations lacking wt multimer staining although defined functions can be observed. We hope that the data including also the novel experiments convinces the reviewer that in-depth characterization of neoantigen-specific T cells has been performed providing extensive information about the specificity and functional quality of defined T-cell responses.

Reviewer #2 : Expert in Antigen proteomic
(Remarks to the Author):

A. Summary of the key results

The manuscript by Bassani-Sternberg et al. explores the immunopeptidomes of melanoma biopsies and provides a rich data set from which to examine the presence of tumour specific peptides, although it should be noted that healthy tissue was not included in the analysis and therefore the absolute tumour specificity is imputed from exome sequencing.

We have provided data for the immunopeptidome of 25 melanoma patients and certainly most of the peptide ligands are not tumor-specific. We hope that this becomes clear in the text. In case of metastatic melanoma, there is no standard healthy tissue to be reasonably used as a control as native melanocytes cannot be removed from healthy donors in sufficient amounts for HLAp analysis. However, the high amount of melanoma-associated antigens identified in these samples clearly demonstrates high tissue specificity of this approach and such melanoma antigens were not identified in other immunopeptidomics dataset we have

analyzed before (2, 3).

Indeed, tumor-restricted neoantigens were identified on the base of exome sequencing, which was performed on the tumor and autologous healthy matched tissue represented in this study by PBMC.

In addition to finding peptides from known tumour antigens such as CT-antigens, phosphopeptides that may represent dysregulated signalling in the cancer cells and mutated peptides associated with tumour specific mutations which could play a role in tumorigenesis were identified.

B. Originality and interest: if not novel, please give references

There is no doubt this is a hot area of research and although overplayed in the text this is one of the first studies to identify mutated peptides/neopeptides. In addition the immunogenicity of such peptides was also examined.

C. Data & methodology: validity of approach, quality of data, quality of presentation

The approach is excellent and data quality is high as one might expect from such high profile laboratories.

D. Appropriate use of statistics and treatment of uncertainties

This is OK

E. Conclusions: robustness, validity, reliability

Overall the conclusions of this manuscript are valid, although the discussion is poorly written and difficult to follow in places. A thorough edit of the text is required to aid clarity and improve grammatical errors.

We revised the text accordingly and hope that this improved clarity.

The statement (p16 lns 289-292) that immunopeptidomics studies sample trypsin sensitive sites is not substantiated and should be removed or altered to clearly indicate this is speculation

We modified this sentence accordingly.

Some details regarding the patient therapy and background would help understand some arguments. The anti-CTLA-4 therapy facilitates activation of T cells presumably through natural priming from tumour derived antigens. This is not indicated and for those not in the immunoncology field this argument should be clearly defined and appropriate background given. This is the same issue with the discussion of PMEL immunised patients in previous studies, not enough background to allow the reader to reach the same

conclusion as the authors.

We have added more patient information (Supplementary Fig. S3 and Fig. 5a) and clarified these issues.

F. Suggested improvements: experiments, data for possible revision

1) Edit of text to improve readability

We have modified the text accordingly.

2) Details of the amount of tumour biopsy used for the peptide elutions should be included and more details regarding its preparation prior to IP provided.

We have included more information in the methods section as follows:

The amount of tumor material that has been available for this research varied significantly, from about 0.1g to 4x1g in Mel15. For the purification of HLA complexes, snap-frozen melanoma tissue samples were homogenized for 10 s on ice using ULTRA-TURRAX (IKA, Staufen, Germany) in a tube containing 5 to 10 ml of lysis buffer and incubated at 4 °C for 1 h. The lysis buffer contained 0.25% sodium deoxycholate, 0.2 mM iodoacetamide, 1 mM EDTA, 1:200 Protease Inhibitors Cocktail (Sigma-Aldrich, MO), 1 mM PMSF, 1% octyl- β -D glucopyranoside (Sigma-Aldrich, MO) in PBS.

3) What do the authors think the peptides that bind to both class I and class II represent? Have they formally shown these are not simply contaminants or copurifying class I/classII IP'd with the reciprocal mAbs? Binding studies showing class II and class I binding would be reassuring otherwise I am not convinced this is a justifiable category.

We thank the reviewer for this very important comment. We have indeed detected similar peptides in both the HLA class I and class II IP experiments and we decided to report this result as it is, without filtering out any of these peptides. We modified the text to better explain that the peptides were actually only detected by MS after both IPs. Several reasons persuaded us to report those overlapping class I class II peptides, as a large proportion of them might indeed be true ligands: 1. With our methodology, more than 95% of the peptides fit to distinct HLA allotypes (2), supporting the yield and purity of the eluted peptides. 2. Among the long HLA-I peptides, many of these seem to fit very well to the corresponding motifs as shown in Fig. 1b. and 1c. 3. Others also reported about the identification of long HLA-I binding peptides (4-6). We hypothesize, that identical peptides that have been detected in both, class I and II peptidome, may be related to common cellular processing, as supported by many previous studies regarding cross presentation and alternative processing pathways of antigens, especially in dendritic cells (7-9). For example, presentation of endogenous

proteins on class II peptides that is dependent on TAP and proteasomal degradation and alternative class I presentation pathways that are dependent on low pH, related to phagosome-endosomes and to recycling of HLA molecules, also of exogenously loaded proteins. Therefore the possibility that the exact peptide sequences will be presented on both class I and class II complexes is biologically feasible. We included these aspects into the discussion.

With respect to the technical aspects, we have previously confirmed in the lab with Western Blot analyses of eluted complexes (unpublished data) that there is no cross reactivity between HB145 and W6-32 antibodies. Class I peptides have been purified before class II peptides, reducing the chance of cross contamination of antibodies. We have data from different cell lines that support this observation, also when the IPs were done in the reciprocal order or separately only for class I and class II, again, no HLA-I molecules are present in the HLA-II IPs and vice versa. Therefore we are convinced that some overlap between class I and class II peptides actually exists. There is indeed a chance that among these long peptides there are also shared contaminants, however, their contribution is likely to be minor. Binding assays for such large collection of peptides is beyond the scope of this study.

4) The failure to be able to predict accurately the affinity of peptides bound to HLA allomorphs with little training data is a valid point but it begs the question is Fig 1d and e necessary?

The purpose of Fig. 1d and 1e is to show that applying a 500nM threshold for predicting binders will not produce similar results in all HLA types, and instead we propose to use the ranked 2% filter. As prediction algorithms have been trained mostly with data from in-vitro binding assays, there are consequently additional biases (related to the inherent dependency of some HLA types on cellular chaperons for peptide loading, for example). We confirmed with our in-depth accurate dataset that the 500nM threshold is in general sufficient, but it is not applicable to all allotypes. This also explains why only 8 potential neo-antigens were predicted for this HLA type while the eluted peptides that bind this HLA molecule are well represented in our IPs.

5) In Fig 2d it is unclear what is being examined and how the peptide abundance was normalised? Or does this represent the number of ligands from a given antigen? If the latter what about peptide abundance?

In Fig. 2 d-g the Y-axis represents the number of unique peptides identified from PMEL divided by the number of total peptides

identified in each sample. As we have shown before in MS-based immunopeptidomics studies, presentation of HLA-I peptides that is measured by the MS intensity does not correlate with expression level of the source proteins, however peptide count does (2). Intensity signals derived from the MS analyses of HLA peptides (even from a given protein) vary in several orders of magnitude and poses technical issues and biases that are related to their propensity to be detected by MS, their binding affinity to the different HLA allotypes, stability of the complexes and cellular processing etc. T cells are sensitive to a very low copy number of peptides. Therefore, a broad repertoire is probably a better estimation of the magnitude of presentation than the sum of measured MS intensities of different peptide sequences.

This has now been stated more clearly in the text. It is clear to the authors that other factors including biochemical properties of the peptides may still have an impact on the number of peptides derived from defined antigens.

6) P.11 In 213 - it is not clear what hotspot refers to?

We have modified the text in the discussion to better explain the concept of hotspots.

7) A more thorough explanation of the T cell assays in the text is required. It is hard to follow what data represents direct ex vivo analysis using ELIspot and what represents in vitro expanded T cell specificities. Please provide a more in depth narrative.

We reorganized the data, added additional data as well as information with respect to the T-cell assays. We also reshaped the text and hope that information and clarity has now improved.

8) P17 In 329 - which self-antigens are you referring to?

We exchanged the word self antigen by non-mutated peptide ligands

G. References: appropriate credit to previous work?

n/a

H. Clarity and context: lucidity of abstract/summary, appropriateness of abstract, introduction and conclusions

See above

See above

Reviewer #3 : Expert in melanoma immunotherapy
(Remarks to the Author):

This is a report of a technological improvement of cancer associated antigen-derived peptide identification using tandem mass spectrometry. The experiments are meaningful and well-controlled. The ability to identify peptides encoded by mutated genes is important and may help the development of novel cancer immunotherapies and biomarkers. However, the current study focuses on methodological advancement, without applying the new ability to identify mutated and phosphopeptides, limiting its overall impact. In essence, this is a methodological advancement. The detection of NCAPG2 and SYTL4-specific T cells, for example, is heartening, but it is not clear that these T cells were involved in the patient's disease course.

We do not agree with this overall estimation. We have provided a comprehensive analysis of autologous and allogeneic matched T-cell responses against newly identified neoantigens which has been now further accomplished (please also see comment in response to reviewer 1). Mainly recently, a number of publications reported on neoantigen-specific T-cell responses of autologous patients as well as matched healthy donors partially just published during the review process of this manuscript (10-14). All of them rather demonstrate case reports and do not provide a broad overview of neoantigen-specific T-cell responses over a broader patient population. However, the information of these cases contributes enormously to our understanding of neoantigen-specific immune responses. We here show a highly comprehensive functional immunomonitoring including proof of neoantigen-specific T cells in the peripheral blood and tumor-infiltrating lymphocytes, neoantigen-specific T cells recognizing freshly isolated tumor cells as well as endogenously processed peptides after minigene transfer, functional avidity of neoantigen-specific T cells and neoantigen-specific T cells with cytotoxic activity. Moreover, our data show very distinct features highly relevant for the development of cancer immunotherapies and biomarkers and are therefore of great interest for the community:

- The number validated mutated peptide ligands recognized by the patient's T cells was high (2 out of eight) when compared to the prediction approach providing hits mainly in the lower single digit percentage range (10, 11, 15-17). Thus, the immunological data was essential to validate the novel technology and demonstrates that this new technology provides the scope to shorten time and efforts to identify neoantigen-specific immune responses.

- We are able to detect neoantigen-specific T cells from unselected PBMC as early as two days after peptide-specific stimulation pointing again to a major role of these antigens. Other reports base on enrichment by multimers or PD-1+ populations (10, 11).
- We have shown that neoantigen-specific responses can be tracked over a period of almost a year during the course of disease and that this correlates well with the clinical course in the selected patient. Thus, these data indicate that such neoantigen-specific T-cell responses may represent highly attractive personalized surrogate biomarkers.
- We demonstrate that multifunctional analyses are necessary to characterize lower avidity T-cell responses against wt peptide. This is highly important for our understanding of autoimmunity associated to novel immunotherapies and further development of adoptive T-cell therapies with respect to safety.

To extend this kind of detailed immune monitoring to a large number of patients is clearly beyond the scope of the current study.

Indeed, the authors show generation of neoepitope-specific T cells from normal donors; it is now widely accepted that T cells against many self and foreign peptides can be generated from normal donors and cancer patients alike.

The idea to isolate neoantigen-specific T cells from a matched donor has not that often been published. There was actually just a manuscript published in the June issue of Science during the review process of our manuscript focusing only on this point (13). In this manuscript, matched T-cell responses were solely isolated with respect to HLA-A2-restricted predicted mutated peptide ligands. We have isolated T cells with specificity for neoantigens presented by diverse HLA types indicating that identification of presented mutated peptide ligands by MS may represent a highly individualized translational approach available also for patients with less common HLA allotypes. Moreover, as stated above, we show that multifunctional analyses are necessary to characterize lower avidity T-cell responses to wt peptides likely highly important for clinical translation.

Similarly, while the authors may be correct that the prevalence of phosphorylation on position 4 of nonapeptides, may mean that these phosphopeptides are attractive targets for immunotherapy across HLA types (line 304), is a speculation, without any evidence in the present manuscript.

We agree that this is a speculation and indicated this accordingly in the text.

Finally, the analysis of T cell responses to neoepitopes in 1 patient in Fig. 5 does not appear to add much beyond what has been previously published on T cell recognition of neoepitopes by other groups.

See above.

Overall, the increased sensitivity of the peptide identification approach by Bassano-Sternberg et al. is an important advancement, but the lack of application of the method to bring forward fundamentally new understanding or application limits the impact of the current study.

1. M. R. Jenkins *et al.*, Visualizing CTL activity for different CD8+ effector T cells supports the idea that lower TCR/epitope avidity may be advantageous for target cell killing. *Cell Death Differ* **16**, 537-542 (2009).
2. M. Bassani-Sternberg, S. Pletscher-Frankild, L. J. Jensen, M. Mann, Mass spectrometry of human leukocyte antigen class I peptidomes reveals strong effects of protein abundance and turnover on antigen presentation. *Molecular & cellular proteomics : MCP* **14**, 658-673 (2015).
3. C. Dargel *et al.*, T Cells Engineered to Express a T-cell Receptor Specific for Glypican Recognize and Kill Hepatoma Cells in Vitro and in Mice. *Gastroenterology*, (2015).
4. R. B. Schittenhelm, N. L. Dudek, N. P. Croft, S. H. Ramarathinam, A. W. Purcell, A comprehensive analysis of constitutive naturally processed and presented HLA-C*04:01 (Cw4)-specific peptides. *Tissue Antigens* **83**, 174-179 (2014).
5. T. Trolle *et al.*, The Length Distribution of Class I-Restricted T Cell Epitopes Is Determined by Both Peptide Supply and MHC Allele-Specific Binding Preference. *Journal of immunology* **196**, 1480-1487 (2016).
6. C. McMurtrey *et al.*, Toxoplasma gondii peptide ligands open the gate of the HLA class I binding groove. *Elife* **5**, (2016).
7. A. Rodriguez, A. Regnault, M. Kleijmeer, P. Ricciardi-Castagnoli, S. Amigorena, Selective transport of internalized antigens to the cytosol for MHC class I presentation in dendritic cells. *Nat Cell Biol* **1**, 362-368 (1999).
8. S. Apcher, R. Prado Martins, R. Fahraeus, The source of MHC class I presented peptides and its implications. *Current opinion in immunology* **40**, 117-122 (2016).
9. A. L. Ackerman, P. Cresswell, Cellular mechanisms governing cross-presentation of exogenous antigens. *Nature immunology* **5**, 678-684 (2004).
10. C. J. Cohen *et al.*, Isolation of neoantigen-specific T cells from tumor and peripheral lymphocytes. *The Journal of clinical investigation* **125**, 3981-3991 (2015).
11. A. Gros *et al.*, Prospective identification of neoantigen-specific lymphocytes in the peripheral blood of melanoma patients. *Nature medicine* **22**, 433-438 (2016).
12. E. M. Verdegaal *et al.*, Neoantigen landscape dynamics during human melanoma-T cell interactions. *Nature*, (2016).
13. E. Stronen *et al.*, Targeting of cancer neoantigens with donor-derived T cell receptor repertoires. *Science* **352**, 1337-1341 (2016).
14. N. van Rooij *et al.*, Tumor Exome Analysis Reveals Neoantigen-Specific T-Cell Reactivity in an Ipilimumab-Responsive Melanoma. *Journal of clinical oncology : official journal of the American Society of Clinical Oncology*, (2013).

15. C. Linnemann *et al.*, High-throughput epitope discovery reveals frequent recognition of neo-antigens by CD4⁺ T cells in human melanoma. *Nature medicine* **21**, 81-85 (2015).
16. N. McGranahan *et al.*, Clonal neoantigens elicit T cell immunoreactivity and sensitivity to immune checkpoint blockade. *Science*, (2016).
17. E. Tran *et al.*, Immunogenicity of somatic mutations in human gastrointestinal cancers. *Science* **350**, 1387-1390 (2015).

REVIEWERS' COMMENTS:

Reviewer #1 (Remarks to the Author):

The authors have addressed my comments satisfactorily.

Reviewer #2 (Remarks to the Author):

The revised manuscript is much improved and the responses to my original criticisms were satisfactory.

Reviewer #3 (Remarks to the Author):

The authors have effectively addressed this reviewer's concerns

Point-by-point-reply

We thank the reviewers for their positive response. We are very pleased that they found all issues resolved.

REVIEWERS' COMMENTS:

Reviewer #1 (Remarks to the Author):

The authors have addressed my comments satisfactorily.

Reviewer #2 (Remarks to the Author):

The revised manuscript is much improved and the responses to my original criticisms were satisfactory.

Reviewer #3 (Remarks to the Author):

The authors have effectively addressed this reviewer's concerns